# NeuroBOLT: Resting-state EEG-to-fMRI Synthesis with Multi-dimensional Feature Mapping

**Yamin Li**[1]   **Ange Lou**[1]   **Ziyuan Xu**[1]   **Shengchao Zhang**[1]   **Shiyu Wang**[1]
**Dario J. Englot**[2]   **Soheil Kolouri**[1]   **Daniel Moyer**[1]   **Roza G. Bayrak**[1]   **Catie Chang**[1]
[1]Vanderbilt University    [2]Vanderbilt University Medical Center
yamin.li@vanderbilt.edu

## Abstract

Functional magnetic resonance imaging (fMRI) is an indispensable tool in modern neuroscience, providing a non-invasive window into whole-brain dynamics at millimeter-scale spatial resolution. However, fMRI is constrained by issues such as high operation costs and immobility. With the rapid advancements in cross-modality synthesis and brain decoding, the use of deep neural networks has emerged as a promising solution for inferring whole-brain, high-resolution fMRI features directly from electroencephalography (EEG), a more widely accessible and portable neuroimaging modality. Nonetheless, the complex projection from neural activity to fMRI hemodynamic responses and the spatial ambiguity of EEG pose substantial challenges both in modeling and interpretability. Relatively few studies to date have developed approaches for EEG-fMRI translation, and although they have made significant strides, the inference of fMRI signals in a given study has been limited to a small set of brain areas and to a single condition (i.e., either resting-state or a specific task). The capability to predict fMRI signals in other brain areas, as well as to generalize across conditions, remain critical gaps in the field. To tackle these challenges, we introduce a novel and generalizable framework: **NeuroBOLT**[1], i.e., **Neuro**-to-**BOL**D **T**ransformer, which leverages multi-dimensional representation learning from temporal, spatial, and spectral domains to translate raw EEG data to the corresponding fMRI activity signals across the brain. Our experiments demonstrate that NeuroBOLT effectively reconstructs unseen resting-state fMRI signals from primary sensory, high-level cognitive areas, and deep subcortical brain regions, achieving state-of-the-art accuracy with the potential to generalize across varying conditions and sites, which significantly advances the integration of these two modalities.

## 1   Introduction

Functional magnetic resonance imaging (fMRI) and electroencephalography (EEG) are the most commonly utilized non-invasive neuroimaging techniques, providing crucial insights into brain functionality. These two modalities offer distinct advantages and limitations that can complement one another when combined [45, 13]. fMRI offers high spatial resolution imaging of whole-brain activity by measuring blood-oxygen-level-dependent (BOLD) signal changes, which facilitates probing regional and network-level function. However, fMRI suffers from low temporal resolution and hemodynamic blurring, which limits its ability to capture the accurate timing of rapid neuronal activity dynamics. Additionally, its high cost, non-portability, and incompatibility with metal implants further restrict the utility of MRI in certain contexts. Conversely, EEG stands out as a low-cost, portable neuroimaging modality with high temporal resolution [10, 30, 61]. However, EEG faces

---

[1]Project page: https://soupeeli.github.io/NeuroBOLT

38th Conference on Neural Information Processing Systems (NeurIPS 2024).

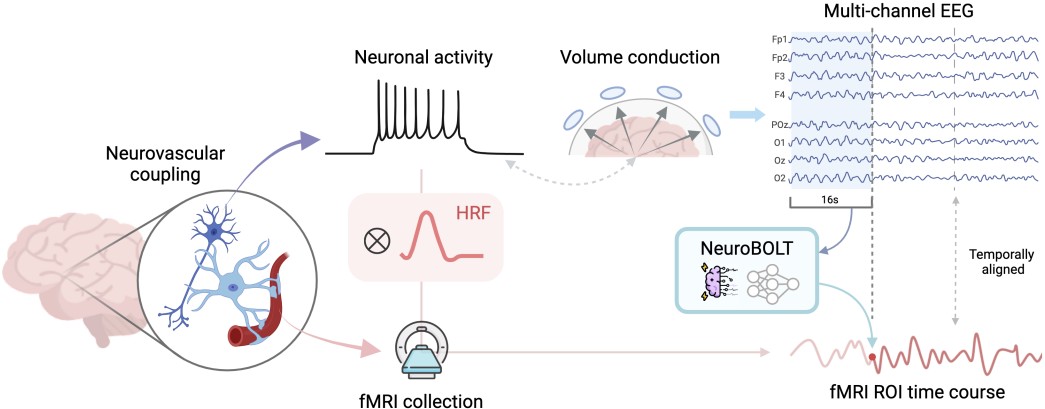

Figure 1: **Overall illustration of EEG-to-BOLD fMRI translation using NeuroBOLT.**

limitations due to volume conduction and from the superficial location of electrodes (on the scalp), making it difficult for EEG to accurately infer the origins of the measured electrical potentials. In this context, the high spatial resolution of fMRI becomes indispensable, especially for imaging deep brain regions.

One approach for investigating the relationship between EEG and fMRI signals is by analyzing data collected simultaneously from both modalities. Such studies have demonstrated correlations between fMRI data and various features of EEG signals, such as the power in certain frequency bands [27, 7, 28], which underscore the potential of EEG to inform fMRI features. However, factors such as the disparate biophysical origins of the two signals, and differences in their spatial and temporal resolutions, can limit the accuracy, interpretation, and consistency of correlation-based EEG-fMRI studies. These challenges are exacerbated in conditions during which a participant is resting passively (i.e., resting state), which is characterized by significant noise and randomness. Consequently, the mechanisms linking neuronal activity to BOLD signals remain only partially understood, posing challenges in mapping EEG to fMRI.

With recent progress in cross-modality synthesis and brain decoding techniques using deep neural networks [8, 9, 16], EEG-to-fMRI synthesis has emerged as a promising yet largely untapped research area [4]. Early pioneering work by [39, 43] employed ridge regression on EEG temporal-spectral features to reconstruct fMRI signals from the visual cortex and sub-cortical regions. More recent studies have used deep neural networks for EEG-fMRI translation [33–35, 5, 3, 25, 31], and those of [25] and [31] developed sequence-to-sequence (Seq-to-Seq) models to reconstruct fMRI time series of deep brain regions from EEG. However, these efforts have primarily focused on task datasets, including cued eye opening/closing [25, 31, 34, 35, 54, 3], leaving the (fully eyes-closed) resting state condition largely unexplored. In addition, prior studies rely on a subject-specific approach, wherein models are solely trained and tested on different sections of the same individual's scans. Further, only a small number of brain regions have been examined in current studies, leaving as an open question the predictive power of EEG for fMRI signals across broader areas (see Appendix B for further discussion of related work). These limitations underscore the need for developing more generalizable models for EEG-fMRI translation that are accompanied by more comprehensive evaluation.

In this paper, we present NeuroBOLT: a multi-dimensional transformer-based EEG encoding framework for Neural-to-BOLD fMRI Translation (Figure 2). NeuroBOLT is designed to capture and integrate multi-dimensional representations across temporal, spatial, and spectral domains, offering a generalizable approach for translating raw EEG waves to BOLD fMRI time series from regions of interest (ROIs). Drawing inspiration from vision transformer (ViT) [15] and its adaptation for multi-channel biosignals [59, 22], our framework transforms raw EEG data into unified "EEG patches", enabling flexible and effective EEG data encoding. These EEG patches are then passed into two parallel sub-modules: (1) spatiotemporal and (2) spectral representation learning modules. The spatiotemporal module is propelled by a recently proposed EEG foundation model (LaBraM) [22], which has been trained to learn effective representations on about 2,500 hours of various types of EEG signals. Moreover, as previous studies have emphasized the importance of leveraging spectral features in EEG representation learning [59, 57, 60], EEG-fMRI correlations [27, 14], and EEG-to-fMRI synthesis [31, 39, 25], the spectral module incorporates multi-scale encoding on EEG spectrogram patches. Specifically, instead of employing a fixed window size for the Short-Time Fourier Transform

(STFT) as commonly used [59, 58, 24, 11], we incorporate spectral features from windows of varying scales. This approach retains the advantages of high temporal resolution from smaller windows and high frequency resolution from larger windows, thereby enhancing the spectral analysis. The output embeddings from the above two encoding modules are then integrated, allowing NeuroBOLT to capture the complexity of neural dynamics and learn the projection from neural to BOLD signals. The key contributions of this work are summarized as follows:

1) **Generalizable EEG-to-fMRI translation framework.** We introduce a novel approach, NeuroBOLT, that utilizes multi-spectral representation to predict high-dimensional fMRI signals from raw EEG data without relying on predefined assumptions about the hemodynamic delay between fMRI and EEG signals. Additionally, our model is designed to accommodate any number of EEG channels, enhancing its versatility across various experimental setups.

2) **Comprehensive evaluation of the predictive power.** We performed comprehensive experiments on subject-specific and cross-subject learning, across selected ROIs in primary sensory, high-level cognitive, and deep brain regions. Further, we probe the generalizability of our approach across data acquired in both resting state and task (auditory stimulus) conditions.

3) **Successful resting-state fMRI reconstruction.** To our knowledge, this is the first study to successfully reconstruct the eyes-closed resting-state fMRI signal from raw EEG data, with a relatively small number (26) of EEG electrodes.

## 2 Method

### 2.1 Task Formulation

While it is well-established that fMRI signals are coupled to neuronal activity, the details of this process are only partially understood and still under debate [19]. In practice, most studies model the relationship between neuronal activity and fMRI by convolving the assumed neural activity with the hemodynamic response function (HRF), which reaches its peak at about 6 seconds and gradually returns to baseline over the next 12-15 seconds. However, the HRF is not uniform, varying significantly across different brain regions and between individuals. Moreover, although EEG can detect the electrical activity of neural populations at millisecond timescales, it suffers from volume conduction, making it challenging to extract its neural origins. This spatial ambiguity also complicates the interpretation of EEG and its relation to fMRI data.

In this study, we aim to build a neural network to learn this complicated projection from the electrophysiological signal measured by EEG to the BOLD signal measured by fMRI. Our approach tries to overcome the above challenges in the following ways: 1) it captures multi-dimensional features of EEG across temporal, spatial, and spectral domains, learning representations that are crucial for accurate fMRI synthesis; 2) it does not assume a pre-defined delay between the two modalities, instead, we extract time windows of EEG data that span a duration approximating the length of HRF preceding each fMRI data point as the inputs to predict the corresponding fMRI data values (i.e., a Sequence-to-One model), in order to accommodate potential variability across subjects, regions, and frequencies.

Functional MRI data are conventionally represented in four dimensions, and can be expressed as a temporal sequence $S = \{V_1, ..., V_K\}$ with K observations, where each observation is a volume $V \in \mathbb{R}^{x \times y \times z}$ with spatial dimensions $x, y$, and $z$. This high-dimensional fMRI signal is often summarized by representing it as a set of brain areas (i.e., $P$ parcels), each with a corresponding fMRI signal that is averaged over its constituent voxels $v_{x,y,z} \subset V$. Here, we employ such a functional parcellation, Dictionaries of Functional Modes (DiFuMo) [12], which was learned across millions of fMRI volumes (see details in [12, 50]). We utilize the DiFuMo parcellation with $P = 64$ regions, and the resulting parcellated fMRI data $Y \in \mathbb{R}^{P \times K}$ contains the signals from each of these regions.

Since fMRI signals are delayed by several seconds compared to the corresponding neural activity, the relationship between an input EEG window and the corresponding fMRI prediction is defined as $\hat{Y}_{p,t} = f_\theta(X_{t-T:t-1})$, where $\hat{Y}_{p,t} \in \mathbb{R}^1$ is the reconstructed fMRI value from the $p$th ROI at time index $t$, and $X \in \mathbb{R}^{C \times T}$ represents the multi-channel EEG signal inputs with $C$ electrodes, $T$ total timepoints. To estimate $f_\theta(.)$, we formulate an optimization problem that minimizes the loss $\mathcal{L}$

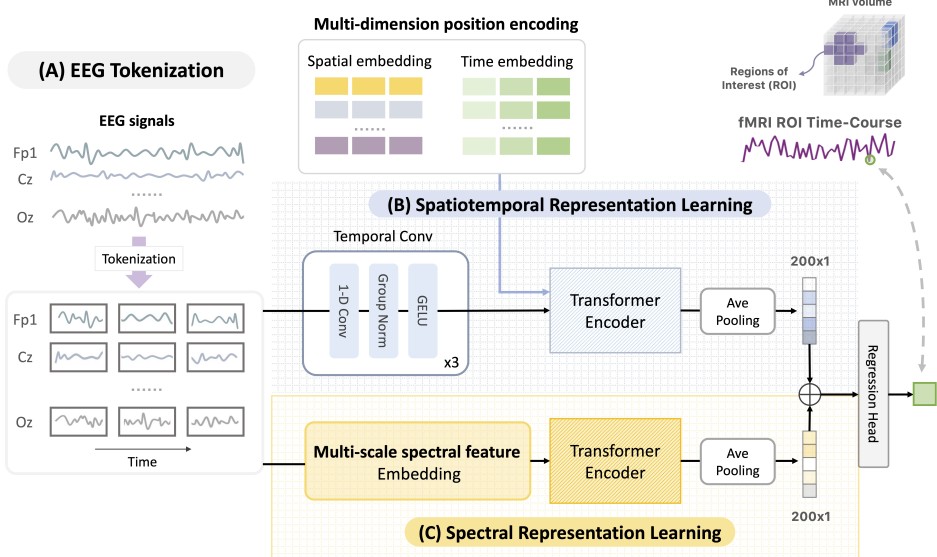

Figure 2: **Overall architecture of NeuroBOLT.** Our method first divides the input EEG window into uniform patches (A), and has two modules that are trained simultaneously: the temporal-spatial representation learning module (B) and the spectral representation learning module (C). The output embeddings from the two modules are summed and used as input to a regression head, which generates the final output.

between the predicted fMRI signal $\hat{Y}_{p,t}$ and the true fMRI signal $Y_{p,t}$ as follows:

$$\min_{f_\theta} \mathbb{E}_X[\mathcal{L}(f_\theta(X_{t-T:t-1}), Y_{p,t})]. \tag{1}$$

The function $f_\theta$ is NeuroBOLT, which includes a temporal-spatial representation module and a spectral representation learning module (Figure 2).

## 2.2 Model Architecture

In this section, we introduce our model: NeuroBOLT, a general architecture for translating EEG signals to fMRI. Our model is designed to accommodate input EEG signals with an arbitrary number of channels. As shown in Figure 2, we leverage the pre-trained EEG foundation model, LaBraM (checkpoint version: LaBraM-base) [22], and finetune on our dataset to obtain spatiotemporal representations of EEG signals. Additionally, we propose multi-scale spectral feature fusion to obtain comprehensive spectral representations. These two modules learn complementary attributes of EEG data. Finally, we integrate the spatiotemporal and multi-scale spectral representations and feed them into a regression head for fMRI prediction.

**Spatiotemporal Representation**  We formulate the multi-channel EEG signals as $X \in \mathbb{R}^{C \times T}$, where $C$ represents the number of EEG electrode channels and $T$ denotes the time length of the input EEG. To obtain the spatiotemporal representation of a given set of EEG signals, we leverage an operation from LaBraM [22], which segments the EEG signals into patches. Assuming the time window length (patch size) for tokenization of EEG signal is $w$ and the stride is $s$, $X$ can be segmented into $\lfloor \frac{T-w}{s} \rfloor + 1$ segments, with each segment denoted as $\mathbf{x} \in \mathbb{R}^{C \times w}$. In this work, we use a window of length $w$ and a stride $s = w$ to segment each EEG channel into patches, obtaining $\mathbf{x} = \{x_{c_j,k} \in \mathbb{R}^w \mid j = 1, 2, \ldots, C, k = 1, 2, \ldots, \lfloor \frac{T}{w} \rfloor\}$. The total number of patches is $C \times \lfloor \frac{T}{w} \rfloor$. These patches are then passed forward to a temporal encoder [22] to obtain the patch embeddings, denoted as:

$$\{e_{c_j,k} \in \mathbb{R}^d \mid j = 1, 2, \ldots, C, k = 1, 2, \ldots, \ldots, \left\lfloor \frac{T}{w} \right\rfloor\} \tag{2}$$

where $d$ is the dimension of the embedding.

To enable the model to capture the temporal and spatial information of the patch embeddings, we set up a list of trainable temporal embeddings and spatial embeddings, denoted as $TE = \{te_k \mid$

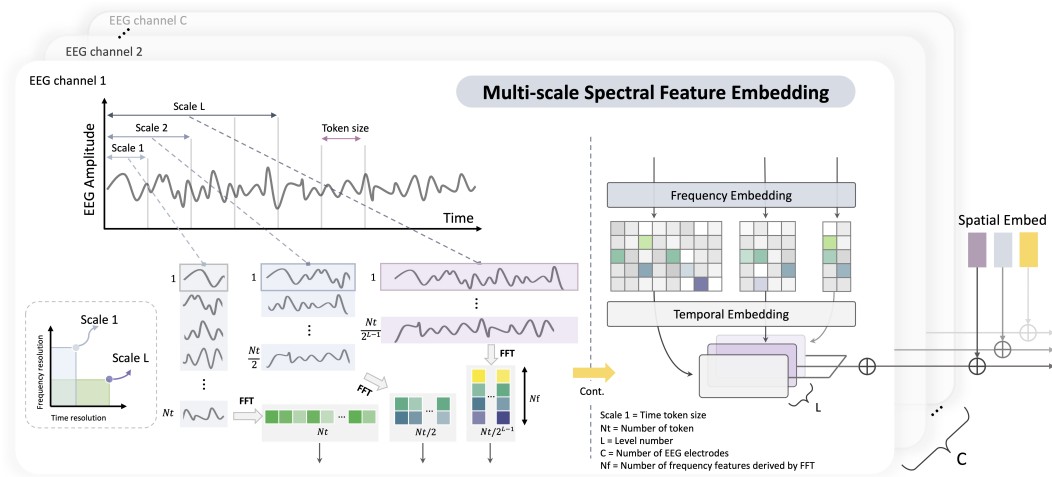

Figure 3: **Multi-scale spectral feature embedding.**

$k = 1, 2, \ldots, \lfloor \frac{T}{w} \rfloor \}$ and $SE = \{se_j \mid j = 1, 2, \ldots, C\}$, respectively. Thus, the final segment embedding $e_{seg}$ can be represented as the sum of the output embedding, temporal embedding, and spatial embedding, denoted as:

$$e_{seg} = \{e_{c_j,k} + te_k + se_j \mid j = 1, 2, \ldots, C, k = 1, 2, \ldots, \left\lfloor \frac{T}{w} \right\rfloor \} \tag{3}$$

The segment embedding $e_{seg}$ is directly fed into the Transformer encoder [52] to obtain the output embeddings. We then apply average pooling to these embeddings to obtain the spatial and temporal representation $r_{st} \in \mathbb{R}^{d_{st}}$, where $d_{st}$ denotes the dimensionality.

**Multi-scale Spectral Representation**   Compared to RGB images, EEG signals present several challenges, such as a low signal-to-noise ratio, apparent stochasticity, nonstationarity, and nonlinear characteristics, making the reconstruction of the original signals difficult [41]. Previous research indicates that the frequency spectrum of EEG signals is crucial for understanding the brain's neurophysiological activities [59, 55]. Therefore, in our work, we utilize the Short-Time Fourier Transform (STFT) to achieve a spectral representation of EEG signals. Unlike most recent state-of-the-art methods, we design a multi-scale spectral approach that captures both coarse and fine representations in the temporal and frequency domains [56, 36]. Details of rationale can be found in Appendix C). For the EEG signals $X \in \mathbb{R}^{C \times T}$, we set the base window size as $w_b$ and define the window size at level $l$ as $w_l = w_b \times 2^l$, where $l = 0, 1, \ldots, L$ represents the level number. Thus, the EEG patches with different window size can be denoted as $\mathbf{x}_l = \{x_{l,c_j,k} \in \mathbb{R}^{w_l} \mid j = 1, 2, \ldots, C, k = 1, 2, \ldots, \left\lfloor \frac{T}{w_l} \right\rfloor \}$. After obtaining the EEG patches, Fast Fourier Transform (FFT) is applied to each patch with the FFT size matches the window length to calculate the magnitude spectrum with $\frac{w_l}{2} + 1$ frequency bins at each level $l$, which can be denoted as $s_l \in \mathbb{R}^{C \times \lfloor \frac{T}{w_l} \rfloor \times (\frac{w_l}{2} + 1)}$.

Similar to the operation in spatiotemporal module, we define a list of trainable frequency embeddings for each magnitude spectrum at different levels, represented by:

$$FE = \{fe_l \in \mathbb{R}^{C \times \lfloor \frac{T}{w_l} \rfloor \times d} \mid l = 0, 1, \ldots, L\} \tag{4}$$

These frequency embeddings, which vary in the number of temporal windows, are further mapped to a set of window embeddings as follows:

$$WE = \{we_l \in \mathbb{R}^{C \times n \times d} \mid l = 0, 1, \ldots, L\} \tag{5}$$

Then, we calculate the sum of the $we_l$ for each level to obtain an overall spectrum embedding as $e_{sp} \in \mathbb{R}^{C \times n \times d}$ as shown in equation 6

$$e_{sp} = \sum_{l=0}^{L} we_l \in \mathbb{R}^{C \times n \times d} \tag{6}$$

For each channel, we apply spatial embedding, similar to the spatiotemporal representation section, to obtain the final embedding with dimensions $e_{sp} \in \mathbb{R}^{C \times n \times d}$, where $C$, $n$ and $d$ represent the channel number, window embedding dimension, and frequency embedding dimension, respectively. The original Transformer is known to have quadratic complexity in both time and space, and EEG signals typically contain tens of channels. To learn these embeddings with lower complexity [59], we feed spectrum embeddings $e_{sp}$ into a linear Transformer Encoder [53, 23]. Let the input embedding $e_{sp} \in \mathbb{R}^{N \times d}$, where $N = C \times n$ represents the number of tokens. The self-attention operation in the linear Transformer Encoder can be formulated as follows:

$$
\begin{aligned}
\mathbf{H} &= Attention(\mathbf{e}_{sp}\mathbf{W}^Q, \mathbf{E}\mathbf{e}_{sp}\mathbf{W}^K, \mathbf{F}\mathbf{e}_{sp}\mathbf{W}^V) \\
&= softmax\underbrace{\left(\frac{(\mathbf{e}_{sp}\mathbf{W}^Q)(\mathbf{E}\mathbf{e}_{sp}\mathbf{W}^K)^\top}{\sqrt{k}}\right)}_{N \times D} \cdot \underbrace{\mathbf{F}\mathbf{e}_{sp}\mathbf{W}^V}_{D \times k}
\end{aligned}
\tag{7}
$$

Here $\mathbf{W}^Q, \mathbf{W}^K, \mathbf{W}^V \in \mathbb{R}^{d \times k}$ are the query, key and value matrices. The self-attention module uses a rank-$D$ approximation for the softmax attention ($N \times N$) by reduced-rank parameter matrices $\mathbf{E}^\top \in \mathbb{R}^{N \times D}$, $\mathbf{F} \in \mathbb{R}^{D \times N}$ (where $D \ll N$). Main components of this module include one linear self-attention layer and one fully connected network. Layer normalization before each component [2], residual connection after each component [18], and dropout right after the self-attention to enable stable training [49, 59]. The output is fed into average pooling along the token dimension to obtain the final spectral representation $r_{sp} \in \mathbb{R}^{d_{sp}}$, where $r_{sp}$ has the same dimension as the spatiotemporal representation $r_{st}$.

**Projection Head**     The hidden embeddings from the above two modules are then summed up and fed into a regression head, which consists of Gaussian Error Linear Unit (GELU) [20] followed by a single linear layer, to make the final prediction of fMRI in the $p^{th}$ ROI at time $t$.

$$
\hat{Y}_{p,t} = \text{Linear}(\text{GELU}(r_{st} + r_{sp}))
\tag{8}
$$

## 3 Experiments

### 3.1 Datasets

**Resting-state dataset**     EEG and fMRI data were collected simultaneously from 24 healthy volunteers in two sessions, each lasting 20 minutes. Scans with significant artifacts in the EEG or fMRI data were excluded for further analysis. The final sample contains 29 fMRI scans from 22 subjects, 7 of whom had two scans. During these fMRI scans, subjects rested passively with their eyes closed (resting state). Written informed consent was obtained, and all protocols were approved by the Institutional Review Board. BOLD fMRI data were collected on a 3T scanner using a multi-echo gradient-echo EPI sequence with repetition time (TR) = 2100 ms. Scalp EEG was acquired simultaneously with fMRI using a 32-channel (10-20 system) MR-compatible system with FCz as the reference (BrainAmps MR, Brain Products GmbH) at a sampling rate of 5 kHz, and was synchronized to the scanner's 10 MHz clock to facilitate MR gradient artifact reduction. Details of EEG and fMRI acquisition, preprocessing as well as artifact reduction can be found in Appendix D.

**Auditory Task Dataset**     To further evaluate the model generalization performance, we also include simultaneous EEG-fMRI data (16 scans from 10 subjects) collected during auditory tasks. Subjects were asked to keep their eyes closed the entire time and to make a right-handed button press as soon as possible upon hearing a tone. This dataset was collected at a different site, different MR scanner (3T Siemens Prisma scanner) and using a slightly different EEG cap (32 channels but with partially different electrode settings). Please also see detailed information about data collection, preprocessing, and experiments in Appendix D.

Unless specified otherwise, the experiments and results below refer to the resting-state data.

### 3.2 Experimental Setup

**Preprocessing**     We employ DiFuMo with $n = 64$ dimensions [12] (see Section 2.1) to extract measured BOLD signals within specific ROIs. We focus on seven ROIs that span a range of spatial

locations and functional roles, namely: **primary sensory regions** (cuneus and Heschl's gyrus), **high-level cognitive areas** (anterior precuneus, anterior and middle frontal gyri), **subcortical regions** (putamen and thalamus), and the **global (whole-brain average) signal**. From these ROI time series, we regress out motion confounds, low-pass filter the signals below 0.15Hz, and use the 95th-percentile of the absolute amplitude to normalize the demeaned ROI time courses. To prepare the EEG data for input, we first exclude the ECG, EOG, and EMG channels (remaining 26 channels), and resample the EEG to 200 Hz to enhance computational efficiency while preserving meaningful frequency components (which are typically below 100 Hz). To predict the fMRI ROI signal, we extract EEG windows of 16 seconds before each fMRI data point. This window length is selected to encompass the peak and most of the variation in the hemodynamic response function. Additionally, this duration aligns with the maximum temporal encoding window length of the pre-trained LaBraM model [22], ensuring optimal fine-tuning. The normalization of the EEG also strictly follows [22], where EEG data values are divided by 100 so that the resulting amplitude falls primarily between -1 to 1, since a typical EEG amplitude range is from -100 $\mu V$ to 100 $\mu V$.

**Baselines and Evaluation Metrics** The baseline models include state-of-the-art EEG encoding models from [59] and [22], as well as two EEG-to-fMRI translation baselines [25, 31]. Since the original EEG-to-fMRI baselines are sequence-to-sequence models, we adapted them by adding a final projection layer to their encoder to enable sequence-to-one prediction. Additional details on baseline implementations are provided in the Appendix E.2. We employed the following two metrics for comparison: 1) **Pearson correlation coefficient (R)**, which measures the strength and direction of the linear relationship between prediction and ground truth; 2) **Mean squared error (MSE)**, which measures the average squared differences between prediction and ground truth across samples. The main text highlights the correlation coefficient as the primary evaluation metric, while MSE results are provided in the Appendix Table 6.

**Implementation Details** For the spatiotemporal module, we initialize the model by loading the pretrained weights from LaBraM-base [22], with a token length = 200, i.e., 1 second, with no overlap. Please refer to [22] for more details of EEG pretraining. For the multi-scale spectral module, we set the smallest scale size $l_0 = 100$, i.e., 0.5 seconds without overlap. Experiments are conducted on a single RTX A5000 GPU using Python 3.9.12, Pytorch 2.0.1, and CUDA 11.7. To ensure consistency during model training, we set a fixed seed across all experiments. The batch sizes are set at 16 and 64 for intra-subject and inter-subject analyses, respectively. AdamW is utilized as our optimizer, and MSE as our training objective. The initial learning rate is set at 3e-4 with a weight decay of 0.05, and a minimal learning rate of 1e-6. For subject-specific prediction, where training and testing occur on the same scan, we split the scan in an 8:1:1 ratio for training, validation, and testing, i.e., training on the first 80% of the data and testing on the last 10%. Given that the fMRI signal exhibits significant autocorrelation, typically extending from about -10 to 10 seconds [6], we implement gaps of 20 seconds between the training and validation sets, as well as between the validation and testing sets, to prevent data leakage. For the inter-subject analysis in resting-state data, we randomly divided the datasets into training/validation/testing sets by approximately 3:1:1 (18 scans : 5 scans : 6 scans). For the task data, we split the scans by 9 scans : 3 scans : 4 scans. Since data from the same individual might have shared representations, the two scans from the same individual are ensured to be in the same set (either training, validation, or testing set) to avoid possible data leakage. All models are optimized on the training set and evaluated on the test set, with the best model and hyperparameters selected based on the validation set.

### 3.3 Intra-subject Prediction

In this section, models were trained on approximately 16 minutes of EEG data, and used to forecast a future interval of fMRI signal (about 2 minutes) within a resting-state scan. Figure 4 illustrates the distribution of correlation coefficients ($R$) between the predictions and the ground truth, along with examples that represent the average performance of our model. The quantitative results are detailed in the upper part of Table 1 and Figure 7 in Appendix. NeuroBOLT outperforms the next two best EEG encoders by 12.26% and 9.71% (respectively) in terms of the average correlation, exhibiting the best performance in reconstructing fMRI signals from all ROIs.

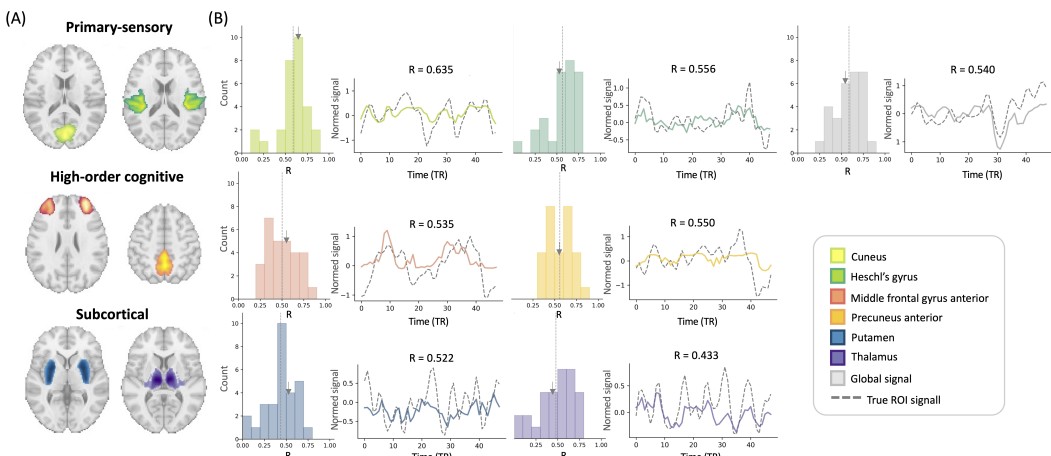

Figure 4: **Intra-subject prediction results.** (A) The parcellation of the chosen ROIs. (B) The distribution of prediction performance (Pearson's correlation coefficients), with example time-series reconstructions that represent performance levels near the mean (indicated by the small grey arrow in the histogram). The dashed lines in the histograms represent mean correlation values.

Table 1: Model performance ($R$) in intra- and inter-subject experiments. **Bold**: the best performance; the underlined: the second-best performance

| | Model | Primary Sensory | | High-level Cognitive | | Subcortical | | - Global Signal | Avg. R↑ |
|---|---|---|---|---|---|---|---|---|---|
| | | Cuneus | Heschl's Gyrus | Middle Frontal | Precuneus Anterior | Putamen | Thalamus | | |
| Intra-scan | BIOT [59] | 0.531±0.223 | 0.518±0.207 | 0.490±0.162 | 0.459±0.110 | 0.410±0.205 | 0.411±0.231 | 0.493±0.133 | 0.473 |
| | LaBraM [22] | 0.540±0.176 | 0.519±0.197 | 0.493±0.153 | 0.490±0.176 | 0.411±0.179 | 0.449±0.177 | 0.487±0.167 | 0.484 |
| | BEIRA [25] | 0.357±0.241 | 0.396±0.240 | 0.294±0.228 | 0.320±0.220 | 0.234±0.194 | 0.328±0.197 | 0.456±0.240 | 0.341 |
| | Li, et al. [31] | 0.460±0.228 | 0.515±0.207 | 0.376±0.169 | 0.457±0.204 | 0.324±0.183 | 0.398±0.194 | 0.583±0.170 | 0.445 |
| | **NeuroBOLT (ours)** | **0.588±0.166** | **0.566±0.183** | **0.502±0.168** | **0.559±0.141** | **0.437±0.184** | **0.480±0.213** | **0.587±0.162** | **0.531** |
| Inter-subject | FFCL [29] | 0.326±0.094 | 0.412±0.039 | 0.327±0.078 | 0.437±0.091 | 0.243±0.125 | 0.373±0.082 | 0.512±0.048 | 0.376 |
| | CNN Transformer [44] | 0.218±0.204 | 0.412±0.114 | 0.298±0.097 | 0.316±0.153 | 0.232±0.086 | 0.180±0.106 | 0.282±0.185 | 0.273 |
| | STT Transformer [48] | 0.269±0.197 | 0.188±0.056 | 0.226±0.130 | 0.280±0.143 | 0.074±0.126 | 0.142±0.101 | 0.347±0.124 | 0.218 |
| | BIOT [59] | 0.457±0.123 | 0.512±0.039 | 0.393±0.128 | 0.445±0.084 | 0.299±0.063 | 0.413±0.073 | 0.529±0.110 | 0.435 |
| | LaBraM [22] | 0.177±0.116 | 0.211±0.105 | 0.153±0.132 | 0.170±0.152 | 0.047±0.111 | 0.147±0.122 | 0.150±0.152 | 0.151 |
| | BEIRA [25] | 0.421±0.112 | 0.482±0.063 | 0.384±0.147 | 0.452±0.149 | 0.241±0.135 | 0.410±0.097 | 0.492±0.106 | 0.412 |
| | Li, et al. [31] | **0.505±0.063** | 0.430±0.048 | 0.415±0.114 | 0.416±0.076 | 0.217±0.139 | 0.424±0.072 | 0.529±0.092 | 0.419 |
| | **NeuroBOLT (ours)** | 0.482±0.100 | **0.561±0.046** | **0.423±0.115** | **0.496±0.136** | **0.335±0.144** | **0.453±0.106** | **0.564±0.115** | **0.473** |

## 3.4 Inter-subject Prediction

**Predicting Resting-state Data from Unseen Subjects** To assess the generalizability of NeuroBOLT across subjects, we first evaluated its performance on held-out recordings in the resting-state dataset, this time using EEG data to predict the fMRI ROI time series across entire scans (with length of about 20 minutes). As shown in the lower part of Table 1, our model achieves the highest average correlation across ROIs, with a score of 0.473. Figure 5 shows the correspondence between the correlation values and the appearance of the reconstructions. This visualization reveals that NeuroBOLT successfully captures major features of fMRI dynamics in reconstructing the whole unseen fMRI scan, especially for primary sensory regions and global signals. Thus, even though both EEG and resting-state fMRI have relatively low signal-to-noise ratio, NeuroBOLT still learns meaningful EEG representations and EEG-fMRI correlations that are crucial for EEG-to-fMRI translation. Additional results including performance distributions with statistical significance, MSE table and examples visualizing the best and the worst fMRI scan reconstructions, are provided in Appendix F for comprehensively displaying the performance. Furthermore, in comparison to other state-of-the-art EEG-fMRI translation approaches [31, 25] and EEG encoding models [59, 22, 29, 44, 48], NeuroBOLT achieves lower mean squared error (MSE) and higher correlation, with at least an 8.74% improvement in average correlations. Similar to the scenario of within-subject prediction, NeuroBOLT shows the highest performance in predicting signals from global and primary sensory regions, achieving average correlations of 0.564 and 0.522, respectively, followed by its performance in high-level cognitive and subcortical regions.

**Ablation Study** In this section, we scrutinize the design of NeuroBOLT on the same unseen resting-state recordings. Table 2 provides an ablation study on the contributions of each component in the NeuroBOLT model. The table reveals that the integration of the spectral learning module markedly

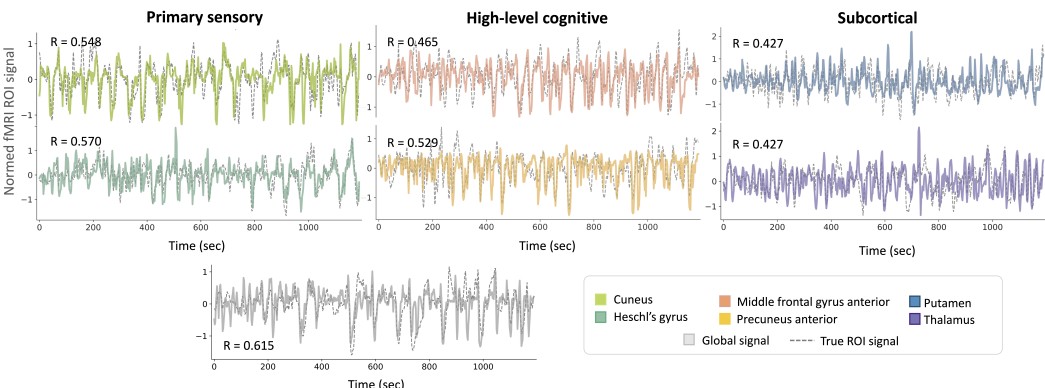

Figure 5: **Examples of reconstruction of the unseen scans.** Scans with the **median** performance are shown. Dashed line: ground truth; colorful lines: model prediction

enhanced the average prediction performance by $0.285$ in $R$, underscoring the pivotal role of spectral features in EEG-fMRI translation. Further, we examine the effectiveness of our multi-scale spectral representation learning module by incrementally increasing the number of included scales. Increasing the number of scales led to significant performance improvements in most of the ROIs.

In our model, the number of FFT points matches the token length. Therefore, larger window sizes produce a spectrogram with higher frequency resolution but at the expense of capturing more rapid changes in temporal dynamics (also see in Figure 3). We observe that fusing multi-scale spectral features leads to better performance in capturing the projection from EEG to fMRI. Additionally, as shown in Table 2, different brain regions appear to prefer various numbers of scales, likely reflecting distinct temporal-spectral dynamics across regions. Overall, the NeuroBOLT configuration with four scales ($l_3$) achieves the best average performance. We have adopted this setting for all of our experiments.

Table 2: Ablation study on the different components of NeuroBOLT and various spectral scale level settings. T represents the Temporal-spatial module, while MS stands for the Multi-scale Spectral module. $l$ denotes the number of scale levels that are used for spectral feature integration, where $l_0$ means that only a single scale that equals the token size is employed.

| Model | Primary Sensory | | | | High-level Cognitive | | | | Subcortical | | | | - | | Avg. R↑ |
| | Cuneus | | Heschl's Gyrus | | Middle Frontal | | Precuneus Anterior | | Putamen | | Thalamus | | Global Signal | | |
| | MSE↓ | R↑ | MSE↓ | R↑ | MSE↓ | R↑ | MSE↓ | R↑ | MSE↓ | R↑ | MSE↓ | R↑ | MSE↓ | R↑ | |
| T [22] | 0.247 | 0.177 | 0.239 | 0.211 | 0.256 | 0.153 | 0.246 | 0.170 | 0.259 | 0.047 | 0.255 | 0.147 | 0.246 | 0.150 | 0.151 |
| MS w/ $l_3$ | 0.208 | 0.404 | 0.193 | 0.486 | 0.222 | 0.384 | 0.193 | 0.489 | 0.245 | 0.274 | 0.222 | 0.452 | 0.181 | 0.534 | 0.432 |
| T+MS w/ $l_0$ | 0.215 | 0.405 | 0.188 | 0.480 | **0.212** | 0.424 | 0.190 | 0.482 | **0.234** | 0.314 | 0.207 | 0.436 | 0.185 | 0.508 | 0.436 |
| T+MS w/ $l_1$ | 0.208 | 0.430 | 0.186 | 0.486 | **0.212** | 0.424 | 0.193 | 0.469 | **0.234** | 0.323 | 0.211 | 0.429 | 0.178 | 0.533 | 0.442 |
| T+MS w/ $l_2$ | 0.197 | 0.464 | 0.174 | 0.542 | 0.213 | **0.432** | 0.189 | 0.491 | 0.235 | 0.317 | 0.215 | 0.433 | 0.177 | 0.549 | 0.461 |
| T+MS w/ $l_3$ | **0.192** | **0.482** | **0.171** | **0.561** | 0.215 | 0.423 | **0.188** | 0.496 | 0.235 | **0.335** | 0.208 | 0.453 | **0.171** | **0.564** | **0.473** |
| T+MS w/ $l_4$ | 0.210 | 0.432 | 0.178 | 0.524 | 0.213 | 0.429 | 0.200 | **0.502** | 0.235 | 0.323 | **0.206** | **0.455** | 0.179 | 0.558 | 0.460 |

**Generalization to Task-related fMRI**    To evaluate the generalizability of our model trained on resting-state data to other conditions, we conducted the following experiments on the task dataset collected from a different site with a different scanner: (1) zero-shot generalization, where we pretrained our model on resting-state data and tested its performance on task data; (2) intra- and inter-subject prediction, where models were trained and evaluated only on task fMRI data; (3) fine-tuning, where models trained on resting-state data were fine-tuned with task fMRI data, and (4) joint-training, where we jointly trained (using both resting-state and auditory task fMRI data) and evaluated the model on the respective held-out test sets of both datasets.

As shown in Table 3, the performance of our pretrained model on zero-shot whole-scan task fMRI reconstruction achieved performances comparable to that of the resting-state data, with even better performance in several regions compared with the model that was trained only on task fMRI. Fine-tuning the model on the task dataset further improved the performance substantially. Moreover, carrying out joint training using both resting-state and auditory task fMRI datasets resulted in the best performance across 4 ROIs in task fMRI prediction. Our results also suggest that joint training is not necessarily facilitating the prediction on resting-state fMRI (beyond training on resting-state data alone), likely due to the smaller sample size of the task dataset and richer variability of brain

dynamics in the resting state condition. More results of task-scan prediction, such as the intra-subject prediction, comparison with baselines, and performance distribution, are provided in Appendix F.4.

Table 3: Performance of NeuroBOLT in inter-subject prediction in resting-state and auditory task fMRI. Mean R values between prediction and g.t. are shown. RS: Resting-State, AT: Auditory Task, RS-p+AT-f: Pretraining on RS and finetuning on AT, RS+AT: joint training of RS and AT.

| Training | Testing | Primary Sensory | | High-level Cognitive | | Subcortical | | - | Avg. R↑ |
|---|---|---|---|---|---|---|---|---|---|
| | | Cuneus | Heschl's Gyrus | Middle Frontal | Precuneus Anterior | Putamen | Thalamus | Global Signal | |
| RS | AT | 0.387±0.087 | 0.431±0.026 | 0.419±0.099 | 0.451±0.050 | 0.240±0.202 | 0.361±0.164 | 0.372±0.087 | 0.380 |
| AT | AT | 0.428±0.141 | 0.479±0.084 | 0.407±0.058 | 0.460±0.071 | 0.187±0.253 | 0.362±0.166 | 0.287±0.120 | 0.373 |
| RS-p+AT-f | AT | 0.446±0.033 | **0.547±0.060** | **0.437±0.089** | 0.471±0.065 | 0.241±0.188 | **0.401±0.177** | 0.385±0.098 | 0.418 |
| RS+AT | AT | **0.461±0.101** | 0.516±0.044 | 0.434±0.106 | **0.476±0.041** | **0.248±0.194** | 0.401±0.220 | **0.404±0.070** | **0.420** |
| RS | RS | **0.482±0.100** | **0.561±0.046** | 0.423±0.115 | **0.496±0.136** | **0.335±0.144** | **0.453±0.106** | **0.564±0.115** | **0.473** |
| RS+AT | RS | 0.478±0.110 | 0.560±0.049 | **0.437±0.086** | 0.494±0.107 | 0.330±0.140 | 0.443±0.074 | 0.540±0.119 | 0.469 |

## 4 Discussion and Conclusion

**Contributions** In this study, we propose NeuroBOLT, a versatile deep-learning solution for projecting scalp EEG to BOLD fMRI signals. By learning a multi-dimensional EEG representation from spatial, temporal, and spectral domains, NeuroBOLT shows a strong capability to reconstruct resting-state fMRI signals from EEG alone. In addition to subject-specific models, it can also predict fMRI time series from held-out subjects across entire 20-minute scans, which is a first in this field. It is important to note that correlations of 0.4-0.5 are considered strong for predicting moment-by-moment signal fluctuations in functional neuroimaging data, given the low signal-to-noise ratio of EEG/fMRI. It is also worth mentioning that our model reconstructs resting-state fMRI signals from deep subcortical regions like the thalamus (with accuracy comparable to, or exceeding, certain cortical ROIs), using raw EEG data with a relatively small set of electrodes (26), enhancing the capabilities of EEG to map subcortical dynamics. This number of channels is typically insufficient to capture detailed neural activity from such regions using methods like EEG source localization [47]. Moreover, NeuroBOLT supports an arbitrary number of EEG input channels, which allows for flexible application across different EEG setups and across different experimental and clinical settings. Overall, by advancing the field of EEG-fMRI translation, this work may ultimately open new possibilities for non-invasive brain research and cost-effective clinical diagnostics.

**Limitations** Currently, our model is trained separately for each brain region, which is inefficient for the ultimate goal of high-resolution fMRI reconstruction. Since functional modules in fMRI also co-fluctuate dynamically, our future work aims to develop an integrated training approach that leverages these co-fluctuations, aiming to enhance the model's ability to reconstruct high-resolution fMRI signals more efficiently. Furthermore, although our model demonstrates promising zero-shot reconstruction performance on unseen task-based scans when trained on resting-state data even with limited samples, the sample size of our resting-state and auditory task dataset remains limited compared to existing studies on single modalities, which may still introduce potential bias or overfitting. For certain comparisons, sizable variability in the performance of the proposed method as well as the baseline methods was also observed (see Appendix F and Figures 7, 8). As our model is adaptable to various channel configurations, we plan to leverage other publicly available simultaneous EEG-fMRI datasets in our future work to train the model on larger, more diverse samples. Additionally, future evaluation with multiple random seeds could help mitigate potential biases especially under small sample size, enhancing the model's reliability and generalizability.

## Acknowledgments and Disclosure of Funding

This work was supported by NIH grants R01 NS112252 and P50 MH109429.

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

# A   Notation Table

Table 4: Notations used in NeuroBOLT

| Symbols | Descriptions |
|---|---|
| $S$ | the 4-D fMRI sequence |
| $V$ | the fMRI volume at each time frame |
| $Y, \hat{Y}$ | the real and predicted fMRI ROI time series |
| $f_\theta(.)$ | the neural network |
| $X$ | the input EEG time sequence |
| $C$ | the number of EEG electrodes |
| $T$ | the time length of input EEG |
| $w$ | the time window length for tokenization of EEG signal |
| $s$ | the stride length |
| $d$ | the dimension of the latent embedding |
| $e_{c_j,k}$ | the embedding of EEG patch at channel $c_j$, $k$th time window in spatiotemporal module |
| $te_k$ | the temporal embeddings of EEG patch at $k$th time window in spatiotemporal module |
| $se_{c_j}$ | the spatial embeddings of EEG patch at channel $c_j$ in spatiotemporal module |
| $TE$ | the temporal embedding set in spatiotemporal module |
| $SE$ | the spatial embedding set in spatiotemporal module |
| $e_{seg}$ | the final EEG patch embedding at certain channel and time window in spatiotemporal module |
| $r_{st}$ | the final representation of spatiotemporal module |
| $d_{st}$ | the dimension of final representation of spatiotemporal module |
| $l$ | the window size level in spectral module |
| $L$ | the number of window size level in spectral module |
| $w_b$ | the base window size in spectral module |
| $w_l$ | the window size at $l$ level in spectral module |
| $fe_l$ | the frequency embedding at level $l$ in spectral module |
| $we_l$ | the window embedding at level $l$ in spectral module |
| $FE$ | the frequency embedding set in spectral module |
| $WE$ | the window embedding set in spectral module |
| $e_{sp}$ | the overall spectrum embedding in spectral module |
| $r_{sp}$ | the final representation of spectral module |
| $d_{sp}$ | the dimension of final representation of spectral module |
| $N$ | the number of tokens |
| $k$ | the new dimensions of tokens after self-attention module |
| $\mathbf{W}^Q, \mathbf{W}^K, \mathbf{W}^V$ | the query, key, and value matrices in self-attention |
| $\mathbf{E}, \mathbf{F}$ | the low-rank projection matrices |
| $\mathbf{H}$ | the output of the self-attention |
| $D$ | the reduced rank of self-attention matrices, $D \ll N$ |

# B   Related Work

Before the adoption of deep learning methods, research into the correlations between EEG and fMRI suggested the potential for inferring high-dimensional fMRI features based on only partial information from relatively low-dimensional EEG measures. Most of these studies investigate EEG-fMRI relationships by regressing the power in one or more EEG frequency bands against the fMRI time series, after convolving the former with a BOLD hemodynamic response model [38, 26]. However, there is currently no comprehensive, definitive model that summarizes the complex translation from EEG to fMRI, possibly owing to complexities such as a potential state-dependence of neurovascular coupling [51], and research into neural-BOLD relationships is still ongoing [21, 19].

Early pioneering work by [39] applied ridge regression on EEG temporal-spectral features to reconstruct fMRI signals from the visual cortex during an eyes-open-eyes-closed task, and to reconstruct fMRI signals from the amygdala during auditory neurofeedback sessions. Since this study attempted to reconstruct fMRI signals only from task-specific brain regions, the ability of EEG to infer fMRI signals from other regions needs further investigation. Using deep neural networks, [33–35, 5, 3, 25, 31]

explored the capability of encoder-decoder based frameworks for EEG-fMRI translation. Specifically, models developed by [33–35, 5] mainly focus on reconstructing fMRI volumes, but their effectiveness in recovering fMRI temporal characteristics has yet to be quantitatively assessed. The work of [25] and [31] addressed this gap in evaluating reconstruction performance in the temporal domain by employing sequence-to-sequence models to directly reconstruct fMRI time series in deep brain regions from raw EEG, analyzing the temporal correlation between predicted and actual fMRI signals. However, these studies are constrained by small sample sizes and rely on a subject-specific approach, wherein models are solely trained and tested on different sections of the same individual's scans. Additionally, most of the existing research has focused on EEG-to-fMRI synthesis during task conditions, including cued eye opening/closing [25, 31, 34, 35, 54, 3], leaving the (fully eyes-closed) resting state condition largely unexplored. This gap in the field may stem from the inherently noisier and more random nature of resting-state data [32], which poses challenges for stable and accurate translation between EEG and fMRI. Yet, resting state is a widely used condition due to its simplicity and ease of applicability in patient populations [37]; therefore, there is a crucial need for models that can successfully operate on resting-state data.

## C Rationale of Designing Multi-scale Spectral Representation

Short-Time Fourier Transform (STFT) is the most widely used method to effectively evaluate how the frequency content of EEG changes over time. Given an multi-channel EEG input sequence $X \in \mathbb{R}^{C \times T}$ with $C$ electrodes and $T$ total time points, $X$ can be segmented into $\lfloor \frac{T-w}{s} \rfloor + 1$ segments. Each segment denoted as $\mathbf{x} \in \mathbb{R}^{C \times w}$, where $s$ represents the stride, and $w$ is the time window length (patch size). Fast Fourier Transform (FFT) is then applied to each segment to calculate the EEG frequency spectrum, where the number of FFT is set equal to the window length $w$. Thus, the number of frequency features derived from each patch $N_f = \frac{w}{2} + 1$. Consequently, the frequency features of EEG sequence input $X$ with stride equal to window length can be denoted as

$$f_{EEG} \in \mathbb{R}^{C \times \lfloor \frac{T}{w} \rfloor \times \lfloor \frac{w}{2}+1 \rfloor} \tag{9}$$

, which serves as the input for spectral representation learning branch.

As noted from above, one of the limitations of the STFT is its fixed resolution. The width of the window affects the representation of the signal, influencing whether frequency resolution (the ability to distinguish closely spaced frequency components) or time resolution (the ability to detect when frequencies change) is prioritized. A wider window $w$ provides better frequency resolution (i.e., higher $N_f$) but results in poorer time resolution (i.e., fewer time windows), while a narrower window enhances time resolution but sacrifices frequency resolution, as illustrated in the bottom left of Figure 3. Moreover, different neural processes are associated with different frequency bands, which may require distinct time windows to capture accurately. Therefore, in this study, we propose a multi-level spectral representation learning approach that captures and aggregates a range of rapid to smooth fluctuations with adaptive windows. By using varying time window lengths, the model can isolate low-frequency, long-duration oscillations with larger windows and high-frequency, short-duration oscillations with smaller windows. This flexibility ensures that each frequency band is represented optimally, enhancing the model's ability to map EEG to fMRI signals accurately as shown in Table 2.

## D Dataset and Preprocessing Details

In this section, we provide more details about the simultaneous EEG-fMRI datasets used in this study as well as the preprocessing steps. The resting-state data included in this analysis were collected from 22 healthy volunteers (mean age $= 35.23 \pm 16.86$, 12 females). During these fMRI scans, subjects were instructed only to rest passively with their eyes closed (resting state), and to try and keep their head still. Participants were compensated 40 dollars. The auditory-task dataset consists of 16 scans from 10 healthy volunteers (mean age $= 27.20 \pm 4.83$, 4 females). During the scans, binaural tones were delivered with randomized inter-stimulus intervals (ISI), and there are two versions of the task that differed only in the timing of tone delivery: (1) a fast ISI version (6 scans, 500 TR/scan, ISI mean $= 5.6 \pm 0.7$ sec); (2) sparse ISI version (10 scans, 693 TR/scan, ISI mean $= 40.1 \pm 14.6$ sec).

Written informed consent was obtained, and all protocols were approved by the Institutional Review Boards of Vanderbilt University and NIH. This non-invasive imaging study poses minimal risk, and individuals with contraindication to MRI were not eligible for participation.

**Removal of Scanner Artifacts in EEG.** For the EEG data, gradient and ballistocardiogram (BCG) artifacts were reduced using BrainVision Analyzer 2 (Brain Products, Munich, Germany), following the parameters and procedures described in more detail in [40, 17]. Gradient artifact reduction was performed using the average artifact subtraction technique[1], utilizing volume triggers. After correcting for gradient artifacts, the EEG data were downsampled to 250 Hz.

**Other EEG Preprocessing Details** The EEG data were collected simultaneously with fMRI using a 32-channel (10-20 system) MR-compatible system with FCz as the reference (BrainAmps MR, Brain Products GmbH). The full channel names for **resting-state** data ordered as below: 'Fp1', 'Fp2', 'F3', 'F4', 'C3', 'C4', 'P3', 'P4', 'O1', 'O2', 'F7', 'F8', 'T7', 'T8', 'P7', 'P8', 'FPz', 'Fz', 'Cz', 'Pz', 'POz', 'Oz', 'FT9', 'FT10', 'TP9', 'TP10', 'EOG1', 'EOG2', 'EMG1', 'EMG2', 'EMG3', 'ECG'. For the **task condition**, the EEG cap channel setting differed slightly: there was no 'FPz', 'FT9', 'FT10', 'EOG1', 'EOG2', 'EMG1', 'EMG2', 'EMG3'. Instead, the cap included the following additional channels: 'FC1', 'FC2', 'CP1', 'CP2', 'FC5', 'FC6', 'CP5', 'CP6'.

The EOG, EMG, and ECG channels are excluded from our analysis, resulting in the final channel number of 26 in resting-state and 31 in task-condition. Before sending into the model, the raw EEG data were first resampled to 200Hz for computational efficiency while maintaining the meaningful frequency components in EEG (typically below 100Hz) according to the Nyquist-Shannon sampling theorem[42, 46], and then normalized by dividing the EEG data values by 100, since the clean EEG data typically fall into the range between -100-100 $\mu V$. In our study, no additional low-pass or high-pass filter is applied to the EEG signal. We also test the model performance in resting-state inter-subject prediction scenario with a high pass filter (0.5Hz) applied to exclude the influence of low-frequency drift. The results with or without filtering are very close, with only a 0.006 absolute difference in average across all regions, which is also non-significant by t-test ($T = 0.1525, p = 0.8814$, two-sided). Therefore, we keep the original settings.

**fMRI Collection and Preprocessing** **Resting-state** MRI data were collected on a 3T Philips scanner. A high-resolution, T1-weighted structural image (TR = 9 ms, TE = 4.60 ms, flip angle = 8 deg, 150 sagittal slices, 1 mm isotropic) was acquired for anatomic reference. BOLD-fMRI data were collected using a multi-echo gradient-echo EPI sequence with a repetition time (TR) = 2100 ms, echo times = 13, 31, and 49 ms, voxel size = 3 × 3 × 3 mm³, slice gap = 1 mm, matrix size = 96 × 96, 30 axial slices. **Task** MRI data were collected on a 3T Siemens Prisma scanner at a different site. The T1-weighted structural images were collected with the following parameters: TR = 2200 ms, TE = 4.25 ms, flip angle = 9 deg, 1 mm isotropic. Multi-echo gradient-echo EPI sequence was also used for collection with TR = 2100 ms, echo times = 13.0, 29.4, and 45.7 ms, voxel size = 3 × 3 × 3 mm³, slice gap = 1 mm, matrix size = 82 × 50, 30 axial slices. For each task scan, the first seven volumes were dropped to allow magnetization to reach steady state.

MRI scanner (volume) triggers were recorded together with the EEG signals for data synchronization. Slice-timing and motion coregistration were applied, followed by noise reduction using multi-echo ICA as implemented in tedana 0.0.9a[2]. Subsequent steps included alignment to an MNI152 standard template (matrix size = 91 × 109 x 91), removal of low-order trends (up to 4th-order polynomials), and spatial smoothing (to 3mm FWHM) using AFNI [3].

**Rationale for ROI Selection** Although our model is designed to predict fMRI signals in any region of the brain, we choose 7 ROIs to report as representative examples that are located in primary sensory, high-level cognitive, and subcortical regions. Cuneus and Heschl's gyrus are responsible for primary visual and auditory processing; the anterior part of the precuneus is particularly involved in several cognitive and perceptual processes; the middle frontal gyrus anterior is part of the frontal lobe, serving an important role in high-level cognitive functions such as working memory, attention, cognitive control, executive function; as for the subcortical regions, the putamen is part of the basal ganglia, serving a crucial role in brain's motor control; thalamus acts as the brain's relay station, transmitting sensory and motor signals to the cerebral cortex. It plays a crucial role in regulating consciousness, sleep, and alertness by processing and relaying information from various sensory systems and the spinal cord to appropriate areas of the brain for further processing.

---

[2]https://tedana.readthedocs.io/en/stable/
[3]https://afni.nimh.nih.gov/afni

Table 5: Hyperparameters for NeuroBOLT

| Hyperparameters | Values |
|---|---|
| Batch size | intra-sub:16, inter-sub:64 |
| Peak learning rate | 3e-4 |
| Minimal learning rate | 1e-6 |
| Learning rate scheduler | Cosine |
| Optimizer | AdamW |
| Adam $\beta$ | (0.9,0.99) |
| Weight decay | 0.05 |
| Total epochs | 30 |
| Warmup epochs | 5 |
| Drop path | 0.1 |
| Layer-wise learning rate decay | 0.65 |

# E   Training Details

## E.1   Hyperparameter Settings

For the spatiotemporal module in NeuroBOLT, we leverage the architecture and pre-trained weights from the state-of-the-art EEG foundation model LaBraM [22] (pre-trained weights version: LaBraM-base), and then fine-tune it on our dataset. To keep consistent with the setting of the pre-trained model, we also use a token size of 1 sec (200 timestamps) without overlap for this module, resulting in 16 tokens per channel. Since we have EEG data with $C = 26$ channels, the final token number would be $16 \times 26 + 1\text{cls token} = 417$ before being sent into the transformer, please see other detailed settings of the model architecture in the "fine-tuning session" of [22].

For the multi-scale frequency spectral representation learning module, we set the smallest scale to 0.5 seconds, corresponding to 100 points at our sampling rate of 200 Hz. For the multi-scale learning, we use $l_3$ throughout our experiments as it achieves the best average performance across other settings, and it includes four window scales: 0.5 seconds, 1 second, 2 seconds, and 4 seconds. The hidden features derived from these four scales are first projected to the same frequency embedding size (equal to the final hidden embedding size: $d_{hiddenembed} = 200$) and then projected to the same temporal embedding shape ($d_{time} = 32$, which is derived by $\frac{T}{scale0}$, where $T = 3200$ is the total input EEG length 3200 timestamps, and as mentioned $scale0 = 100$). These projections are both achieved by using a single layer of Linear projection. Therefore the shape of the embeddings before the transformer module is $(32 \times 26 + 1\text{cls token}, 200)$. In this study, we use a linear transformer[53] instead as our encoder to reduce the complexity and improve the speed. A previous study by [59] proves that the linear and the vanilla transformer yield very similar performance in biosignal representation learning, while the former one takes less time, so we also adopt the transformer with linear complexity in this study, with 8 heads and 4 transformer layers. We obtain the final embedding from each module by an average pooling step. Then the shape of the hidden embeddings from the spatiotemporal and multi-scale spectral modules are summed up, with a final embedding shape $(200, 1)$. Other training details are shown in Table 5.

## E.2   Baselines

To ensure fair comparisons across all baselines, we apply the same seed and a consistent set of hyperparameters - batch size, learning rate, optimizer, and number of training epochs - as detailed in Table 5 for each reconstruction scenario.

**EEG-fMRI Translation:**

- Li et al. [31]: A Encoder-Decoder CNN that leverages effective spectral representation learning with sinusoidal activation function for Sequence-to-Sequence EEG-fMRI prediction in deep brain regions during eyes-open-eyes-closed (EOEC) task. We adapt this model by only using the encoder part and then using linear projection to make predictions similar to our approach.

- BEIRA [25]: An interpretable CNN Encoder-Decoder for deep brain region fMRI reconstruction during EOEC task. We also adapt this model by only using the encoder part and then using linear projection to make predictions similar to our approach.

**EEG Encoding:**

- LaBraM [22]: A recent EEG foundation model trained on over 2,500 hours of EEG data to learn the spatiotemporal information of EEG signals, which achieved the state-of-the-art downstream classification task performance. We maintain the original model architecture with a 200-dimensional latent embedding and utilize the pretrained weights from this model (LaBraM-base, the only publicly available checkpoint for LaBraM) for fine-tuning on our datasets.

- BIOT [59]: A transformer-based flexible bio-signal encoder architecture that learns the temporal-spectral information from EEG signals. Since BIOT and LaBraM share similar input processing, and both produce a one-dimensional latent representation for each sample, we set BIOT's embedding size to match LaBraM and our model ($d_{hiddenembed} = 200$) to ensure a fair comparison, keeping other parameters consistent with the original paper.

- FFCL [29]: A model that combines a CNN model and LSTM model for learning separate representations, which are merged to make the final prediction.

- ST-Transformer [48]: a transformer-based spatial-temporal feature learning neural network for EEG classification task.

- CNN Transformer [44]: A transformer convolutional neural network for automated artifact detection in EEG.

For FFCL, ST-Transformer, and CNN Transformer, we retain the original model architectures as used in the baseline experiments in [59]. For detailed implementation, please refer to their GitHub repository[4].

---

[4]https://github.com/ycq091044/BIOT/tree/main/model

# F    Additional Results

## F.1    Held-out Resting-state Scan Reconstructions Examples

In this section, we show more examples that reflect the best and the worst reconstruction performance over all ROIs in Figure 6. The total length of the scans is 1207.5 seconds. NeuroBOLT shows the capability to reconstruct the whole resting-state fMRI scan over a large time range.

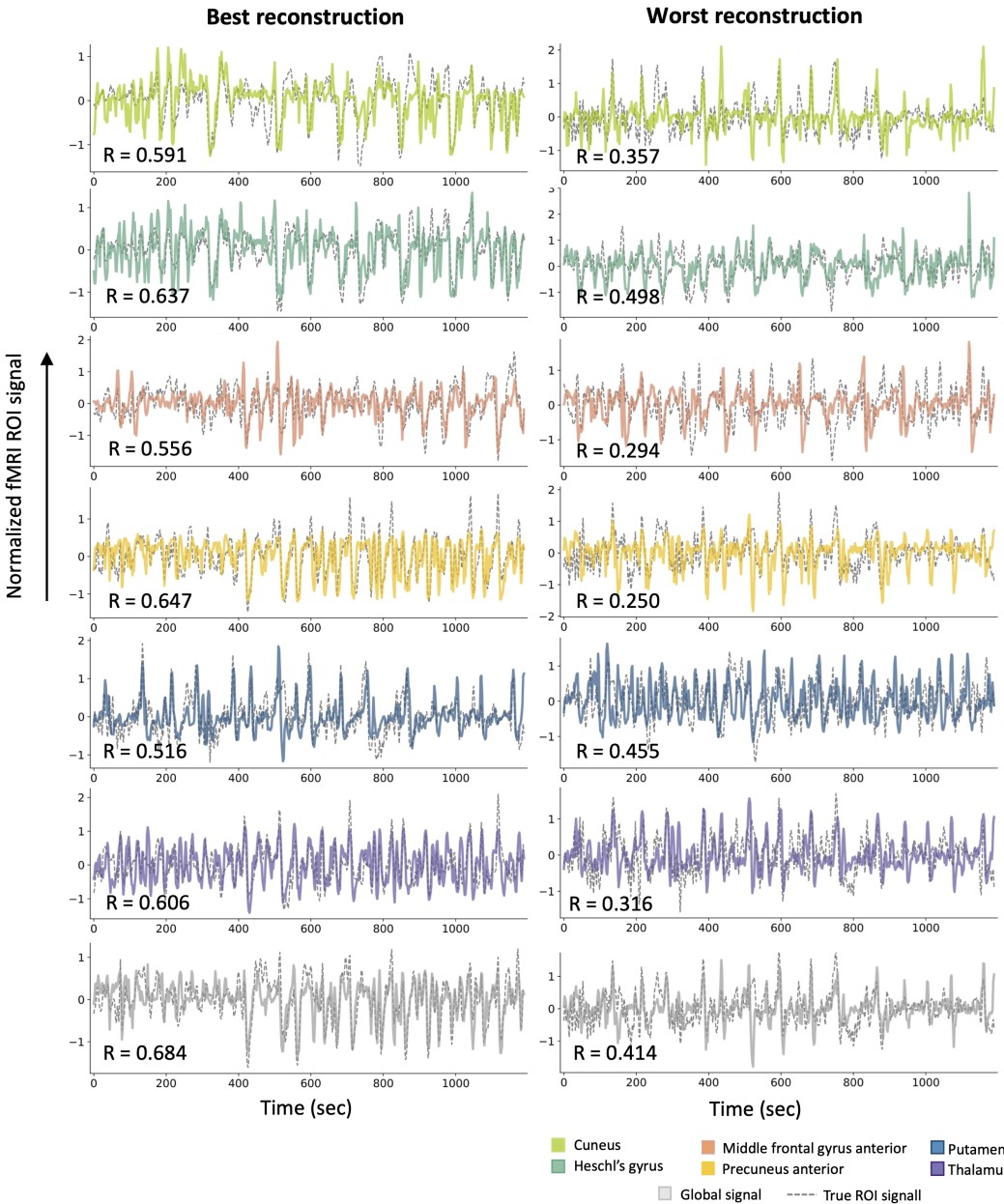

Figure 6: **More reconstruction results for inter-subject prediction (unseen subject)** Normalized predicted and real fMRI ROI signals are shown.

## F.2    MSE Results in Resting-state Data

Here we provide the model performance assessed by MSE values for intra- and inter-subject experiments conducted on resting-state data (see Table 6).

Table 6: MSE in intra- and inter-subject experiments on resting-state data.

| | Model | Primary Sensory | | High-level Cognitive | | Subcortical | | - | Avg. MSE↓ |
|---|---|---|---|---|---|---|---|---|---|
| | | Cuneus | Heschl's Gyrus | Middle Frontal | Precuneus Anterior | Putamen | Thalamus | Global Signal | |
| Intra-scan | BIOT[59] | 0.271 | 0.249 | **0.237** | 0.287 | 0.238 | 0.274 | 0.249 | 0.258 |
| | LaBraM[22] | 0.276 | 0.249 | 0.255 | **0.273** | **0.236** | 0.247 | 0.277 | 0.259 |
| | BEIRA [25] | **0.249** | 0.244 | 0.260 | 0.288 | 0.239 | 0.241 | 0.222 | 0.249 |
| | Li, et al. [31] | 0.261 | **0.226** | 0.275 | 0.275 | 0.249 | 0.229 | 0.226 | 0.249 |
| | **NeuroBOLT (ours)** | 0.272 | 0.242 | 0.263 | 0.274 | 0.238 | **0.225** | **0.221** | **0.248** |
| Inter-subject | FFCL [29] | 0.225 | 0.212 | 0.230 | 0.205 | 0.250 | 0.230 | 0.189 | 0.220 |
| | CNN Transformer [44] | 0.261 | 0.226 | 0.236 | 0.223 | 0.246 | 0.248 | 0.282 | 0.246 |
| | STT Transformer [48] | 0.240 | 0.295 | 0.299 | 0.244 | 0.258 | 0.255 | 0.245 | 0.262 |
| | BIOT [59] | 0.217 | 0.186 | 0.220 | 0.201 | 0.239 | 0.220 | 0.196 | 0.211 |
| | LaBraM [22] | 0.247 | 0.239 | 0.256 | 0.246 | 0.259 | 0.255 | 0.246 | 0.250 |
| | BEIRA [25] | 0.231 | 0.194 | 0.242 | 0.197 | 0.250 | 0.230 | 0.201 | 0.221 |
| | Li, et al. [31] | 0.193 | 0.218 | **0.214** | 0.219 | 0.263 | 0.212 | 0.196 | 0.216 |
| | **NeuroBOLT (ours)** | **0.192** | **0.171** | 0.215 | **0.188** | **0.235** | **0.208** | **0.171** | **0.197** |

## F.3 Distributions of Resting-state Reconstruction Performance

This subsection presents statistical measures for resting-state fMRI scan reconstructions in comparison with other models (see Figure 7). While the presence of statistical significance varies across methods and ROIs, our proposed approach demonstrates consistently higher mean performance than the baselines for all (or all but one) of the ROIs in the intra/inter-scan comparisons.

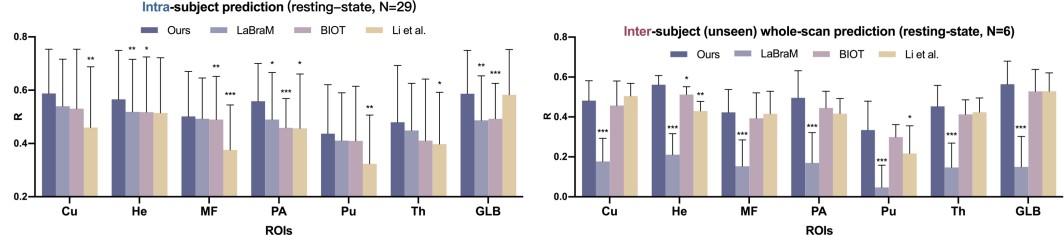

Figure 7: **Intra- and inter-subject EEG-to-fMRI synthesis performance in resting-state.** Paired t-test significance between NeuroBOLT and each baseline: *$p<0.05$, **$p<0.01$, ***$p<0.001$; Error bars are showing the standard deviation (S.D.).

## F.4 Task-fMRI Prediction Performance

In this subsection, we show that, in addition to the resting-state condition, NeuroBOLT is also effective when trained and tested on auditory-task condition in both intra- (Figure 8(A)) and inter-subject conditions (Figure 8(B)), achieving the best unseen scan reconstruction performance in 6 out of 7 ROIs. Figure 8(C) present a comparison example between the unseen task signal reconstructed by the model fully trained on resting-state data (zero-shot) and the signal reconstructed by the model trained on resting-state data and further fine-tuned on task data.

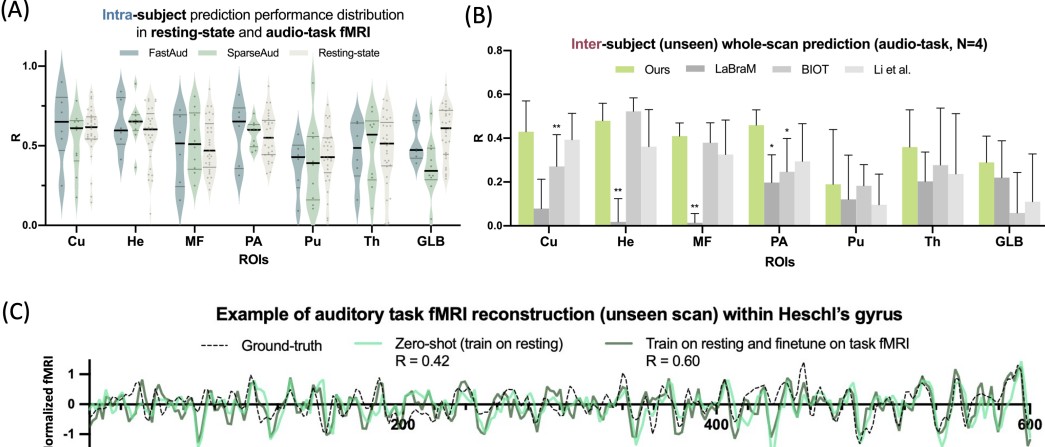

Figure 8: **Task-condition EEG-to-fMRI synthesis performance** (A) Distribution of correlation coefficients between ground truth and signals predicted by NeuroBOLT for intra-subject predictions under resting-state and auditory-task conditions. (B) Inter-subject prediction performance (R value) in 4 held-out auditory task scans compared with other baselines. Paired t-test significance between NeuroBOLT and each baseline: *$p$<0.05, **$p$<0.01, ***$p$<0.001; Error bars are showing the standard deviation (S.D.). (C) Example scan of inter-subject task fMRI reconstruction: zero-shot testing with pretrained model on resting state (shown in light green) and subsequent fine-tuning on task condition (shown in dark green). Note that only partial time series are shown here for better visualization. R values are calculated from full sequences.

