# OpenReview forum: "NeuroBOLT: Resting-state EEG-to-fMRI Synthesis with Multi-dimensional Feature Mapping"
_NeurIPS.cc/2024/Conference — NeurIPS 2024 poster_

### Official Review · Reviewer_bwvZ · 2024-07-11

**Soundness:** 2
**Presentation:** 3
**Contribution:** 2
**Rating:** 4
**Confidence:** 5

**Summary:**

This paper introduces NeuroBOLT, a transformer-based model. NeuroBOLT utilizes multi-dimensional representation learning across temporal, spatial, and spectral domains to translate raw EEG data into comprehensive fMRI activity signals across the entire brain. Experimental results showcase NeuroBOLT's ability to effectively reconstruct resting-state fMRI signals across primary sensory, high-level cognitive areas, and deep subcortical regions.

**Strengths:**

1. The paper tackles one of the most challenging and competitive topics in neuroscience.
2. The motivation behind the paper is quite clear, and the experimental section is logically sound.
3. The figures and tables in the article are clear and well-organized, making it highly readable.

**Weaknesses:**

1. The method abbreviation and the title are not closely related. It is unclear where 'BOLT' comes from in the title, and even after reading the abstract, it remains confusing.
2. In fact, there has been a lot of work on fMRI-EEG in recent years, especially in 2023 and 2024, but the author's related work lacks a significant amount of relevant literature.
3. In the abstract and introduction, the author's description of the method is inconsistent with the organization in the methodology section, resulting in a need for improved readability.
4. The writing of the article needs to be further standardized. For example, 'FMRI data' is sometimes written with a capital 'F' and other times as 'fMRI data'.
5. Although the layout and presentation of the tables are aesthetically pleasing, the font size is too small, making them difficult to read even when enlarged.
6. The paper does not provide code or data to support the reproducibility of results.
7. This paper lacks details on the parameter selection for the baseline methods. Although the authors state, 'The baseline models are from [44] and [16], where we choose the models with the best downstream classification task performance,' the datasets and tasks in references [16] and [44] are not entirely consistent with those in this paper. Therefore, the authors should specify the exact process of parameter selection.
8. The equations and symbols in the article are not very standardized. The authors should provide notation to help readers understand.
9. The authors should conduct statistical tests to validate the significance of their methods.
10. For readers in the NeurIPS community, the theoretical contribution of this paper appears to be weak.
11. I don't quite understand what the author means by the third point of contribution: 'Successful resting-state fMRI reconstruction To our knowledge, this is the first study to successfully reconstruct the resting-state fMRI signal from raw EEG data, with only 26 electrodes.' What is the significance of 26 electrodes?
12. Equation 2 does not appear to be a complete equation.

**Questions:**

Please see the twelve weaknesses above.

**Limitations:**

Limitations are discussed in the section of Discussion and Conclusion.

---

> ### Author Rebuttal · Authors · 2024-08-07
>
> We sincerely appreciate your valuable suggestions, which truly help us enhance the quality and readability of our paper from multiple aspects. Responses to your concerns are presented as follows:
>
> > **Consistency and presentation details (W3-5, W8, W12):**
>
> * **W3**: We restructured the model overview in the intro and abstract, and included summarizing sentences to ensure a coherent and clear presentation of our method throughout the manuscript.
> * **W4**: We have carefully reviewed the manuscript and ensured that we use a capital "F" only when "fMRI" appears at the beginning of a sentence, following standard conventions.
> * **W5**: We have now moved the MSE metric to the appendix to create more space for displaying the mean and standard deviation of R values. We will include an interactive table that allows readers to easily zoom in/out on our project page.
> * **W8**: We have now reviewed all equations and symbols throughout the manuscript to ensure they are standardized. In the revised version of the manuscript, we will include a notation section (table) in the appendix that defines all symbols and terms used in the equations to help readers better understand the mathematical content and ensure consistency throughout the manuscript.
> * **W12**: The expression in question is intended to define the set of patch embeddings (i.e., to define the notation), rather than to present an equation.
>
> > **Meaning of the model abbreviation (W1):**
>
> “NeuroBOLT” stands for “**Neuro**-to-**BOL**D-**T**ranslation.” We have added clarification of this abbreviation both in the abstract and introduction.
>
> > **Related work (W2):**
>
> We have incorporated additional relevant literature and discussed their relevance to our work in the Introduction section of our revised manuscript, including the works from Calhas et al. [1-2] and Liu et al. [3-4].
>
> > **Data and code availability (W6):**
>
> We will release all datasets, code, and model weights upon acceptance. This will provide the necessary resources for the community to validate, further explore, and build upon our models.
>
> > **Training details of baselines (W7):**
>
> We ensured that all training hyperparameters (like batch size) were consistent with those used for our model across all the baselines. Our baseline models consist of (1) SOTA EEG encoders mentioned in [5-6] and (2) EEG-to-fMRI frameworks [7-8], as detailed in Appendix B.2. We strictly adhered to the model settings provided in their original papers when training. We have clarified this and added a table in our revised manuscript that specifies these parameter settings to ensure the results' replicability.
>
> > **Statistical tests (W9):**
>
> We have now added statistical tests for all comparisons with other baseline models (please see **General Response**).
>
> > **Theoretical contribution (W10):**
>
> Our study aims to address a key limitation in previous methods for learning the effective EEG representation useful for EEG-fMRI synthesis. We not only incorporate the SOTA EEG encoding framework but also emphasize the importance of more comprehensive frequency representations, which are often neglected in previous methods [1-4,6-8]. We propose a multi-level spectral representation learning approach that captures and aggregates a range of rapid to smooth fluctuations with adaptive windows. As demonstrated in Table 2 (ablation study), NeuroBOLT benefits from this design by effectively capturing dynamic representations and improving synthesis accuracy and fidelity. Furthermore, additional experiments on an auditory task dataset (please see the **General Response**) validate the **robustness and applicability of our learned representations across neural data from different sites and scanners.**
>
> We believe our approach not only facilitates the translation between two neuroimaging modalities but also highlights the need for multi-scale spectral features for more effective EEG encoding in future research within this community. We will include more detailed theoretical derivations and explanations of spectral representations in the appendix to strengthen the theoretical foundation of our work.
>
> > **Significance of 26 channels (W11):**
>
> For EEG, having fewer than 32 electrodes (26 in our case) is typically considered a small channel set, which presents challenges in accurately estimating neural activity in cortical and subcortical regions using traditional methods like EEG source localization, which often requires >64 channels [9]. However, our experiments show that even with fewer electrodes, we were able to reconstruct the fMRI signals across multiple brain regions in both intra- and inter-subject prediction scenarios. This finding suggests that our method is practical and effective, potentially broadening its applicability to real-world scenarios, such as clinical applications, where electrode count may be limited.
>
> We genuinely appreciate the suggestions, and believe our paper will be improved with your feedback. Please let us know if you have additional questions!
>
> [1] Calhas, David, et al. "Eeg to fmri synthesis for medical decision support: A case study on schizophrenia diagnosis."
>
> [2] Calhas, David, et al. "Eeg to fmri synthesis benefits from attentional graphs of electrode relationships."
>
> [3] Liu, Xueqing, et al. "Latent neural source recovery via transcoding of simultaneous EEG-fMRI."
>
> [4] Liu, Xueqing, et al. "A convolutional neural network for transcoding simultaneously acquired EEG-fMRI data."
>
> [5] Jiang, Wei-Bang, et al. "Large brain model for learning generic representations with tremendous EEG data in BCI."
>
> [6] Yang, Chaoqi, et al. "Biot: Biosignal transformer for cross-data learning in the wild."
>
> [7] Li, Yamin, et al. "Leveraging sinusoidal representation networks to predict fMRI signals from EEG."
>
> [8] Kovalev, Alexander, et al. "fMRI from EEG is only Deep Learning away: the use of interpretable DL to unravel EEG-fMRI relationships."
>
> [9] Michel, Christoph M., et al. "EEG source imaging."

---

> > ### Comment · Reviewer_bwvZ · 2024-08-12
> >
> > Thank you for the author's response.
> >
> > 1. Some specific modifications by the author were not observed, such as W3, W5, W6.
> >
> > 2. Regarding W2, [1-6].... Aside from the paper titled "EEG-to-fMRI Synthesis," many papers titled "simultaneous EEG/FMRI" also align with the direction of this work. Among the articles cited and compared by the author, only 19 and 24 are related to EEG-fMRI translation, while most focus on EEG Encoding. The insufficient comparative experiments also stem from a lack of thorough research in the related work.
> >
> > 3. Regarding W7, the author stated, "We strictly adhered to the model settings provided in their original papers when training." However, the datasets and evaluation metrics used by the author may differ from those in the original paper, which may not be entirely fair.
> >
> > 4. Regarding question 10, the author mentioned "validate the robustness and applicability," but based on the appendix results, it appears that the author did not conduct an in-depth study of robustness, such as the method's robustness under different fMRI and EEG noise levels. Additionally, "We will include more detailed theoretical derivations and explanations of spectral representations in the appendix to strengthen the theoretical foundation of our work." These specific details should also be clarified.
> >
> > [1] Bricman P, Borst J. EEG2fMRI: Cross-Modal Synthesis for Functional Neuroimaging[J]. 2021.
> >
> > [2] Trujillo-Barreto N J, Daunizeau J, Laufs H, et al. EEG–fMRI Information Fusion: Biophysics and Data Analysis[M]//EEG-fMRI: Physiological Basis, Technique, and Applications. Cham: Springer International Publishing, 2023: 695-726.
> >
> > [3] Liu X, Tu T, Sajda P. Inferring latent neural sources via deep transcoding of simultaneously acquired EEG and fMRI[J]. arXiv preprint arXiv:2212.02226, 2022.
> >
> > [4] Wei H, Jafarian A, Zeidman P, et al. Bayesian fusion and multimodal DCM for EEG and fMRI[J]. Neuroimage, 2020, 211: 116595.
> >
> > [5] Calhas D. EEG-to-fMRI Neuroimaging Cross Modal Synthesis in Python[J]. 2023.
> >
> > [6] Tu T, Paisley J, Haufe S, et al. A state-space model for inferring effective connectivity of latent neural dynamics from simultaneous EEG/fMRI[J]. Advances in Neural Information Processing Systems, 2019, 32.
> >
> > .........

---

> ### Author Response · Authors · 2024-08-13
> **Thank you for the comments - Part 1**
>
> Thank you very much for your additional comments. Please note that this year, **authors are not permitted to upload the revised full manuscript or include links to external pages during this rebuttal period.** We were only allowed to upload a one-page PDF (please kindly refer to our [Global Response](https://openreview.net/forum?id=y6qhVtFG77&noteId=VLcizb9WHq)). Therefore, we are unable to provide the revised manuscript here (in response to your **Points 1 and 4.2**), but will upload it when permitted. Responses to your other new concerns are below:
>
>
> >**Point 2:**
>
> We appreciate the reviewers' thorough examination of related work. Our rationale for including particular baselines, and further discussion of your reference list ([1-6]), is provided below:
>
> **Ref. \[2\]** is an excellent review article on simultaneous EEG-fMRI, and we will add it to the other simultaneous EEG-fMRI articles that we have cited in our original manuscript (including Ritter et al., 2006; Laufs et al., 2003, 2006; Chang et al. 2013; de Munck et al., 2009). With regard to the DCM model in **Ref. \[4\]**, we had not selected it as a baseline since it requires specifying stimulus onsets along with the neuroimaging data. Since no stimuli are presented during resting state, this model can not operate on resting-state data without major modifications. In discussing their future directions, the authors of **Ref. \[4\]** state: “Finally, the expansion of the current task-based analysis to the corresponding resting-state methodology, where an equivalent canonical microcircuit formulation for cross spectral data features, will be needed.” **Ref. \[6\]**, aims to infer the effective (directed) connectivity of a brain network based on the complementary information provided by EEG and fMRI. This is interesting work too, but does not appear to provide a framework for inferring fMRI time courses from EEG.
>
> As we mentioned in our first rebuttal, we indeed plan to cite **Ref. \[3\]** (which is the preprint version of *\[3\]* in the previous response) and **Ref. \[5\]** (which uses the same model as *\[1,2\]* in the previous response). For **Ref. \[3\]**, we were unable to find the code, and some experimental settings are not specified in their paper, which would make a direct comparison potentially unfair and unreliable, so we did not include it as a baseline. For **Ref. \[5\]**, the model only accepts 64-channel EEG as input (according to the authors’ GitHub repo and PyPi sites). Since we are using EEG data from 32-channel caps, we did not include this model as a baseline here, but look forward to doing so in future studies with 64-channel data. **Ref \[1\]** explored three conventional architectures: FCNN, CNN, and Transformer. However, the paper did not provide code/model parameters and offered limited information about the training process. In our baselines, we have included models that are similar to or more advanced versions of these architectures (see Appendix B.2 in the original manuscript: CNNs *\[19, 23, 24, 32\]* and Transformers *\[16, 44, 36, 32\]*).
>
> | Baselines | If&nbsp;included | Rationale |
> | :---- | :---- | :---- |
> | [1] | ✗ | - Source code is not available; - Model parameters are not specified; - We already included baselines with similar or advanced versions of these architectures |
> | [3] | ✗ | - Source code is not available; - Model parameters are not specified |
> | [4] | ✗ | Not applicable without major modifications |
> | [5] | ✗ | Only supports 64-channel inputs |
> | [7] | ✓ | Provided in the manuscript. |
> | [8] | ✓ | Provided in the manuscript. |
>
> We hope that our response helps to clarify the rationale behind the baseline models selected for this submission. While the simultaneous EEG-fMRI field is indeed large, the area of EEG-to-fMRI synthesis is currently a niche but rapidly emerging subfield. We will also include the referenced papers in our revised manuscript, which will appear on this forum when we are permitted to submit.
>
> ### **Please see our responses to _<Point 3, Point 4 and references>_ in our [next comment block](https://openreview.net/forum?id=y6qhVtFG77&noteId=wkIJndrOr2)**

---

> ### Author Response · Authors · 2024-08-13
> **Thank you for the comments - Part 2**
>
> ### **Here are the responses to <Point 3, Point 4 and references>** (For responses to **Point 1 and Point 2**, please see the **[comment block above](https://openreview.net/forum?id=y6qhVtFG77&noteId=ssbQ088SCz)**)
>
> >**Point 3:**
>
> If we understand correctly, the reviewer suggests running a grid search for each baseline model to find the parameters that perform optimally on the current dataset and with the current evaluation metrics. In our study, we had adopted the parameter settings recommended by the authors. When evaluating baselines, it is common practice to use the parameters suggested by the original authors, due to the computational burden associated with a comprehensive parameter search across the search space for each application. Moreover, we would like to clarify that we **used the same evaluation metrics as in \[7,8\], i.e., correlation coefficient (i.e., R values).** For the other EEG encoding models, the tasks in the original papers are different (mostly classification tasks like detection of seizures, event type classification, etc.), so metrics like accuracy, AUC, Cohen’s Kappa, and weighted F1 were used in these papers.
>
> To further address your concern regarding differences in dataset, we conducted additional experiments on the ***same dataset*** as in refs. \[7,8\] (eyes open/close task fMRI), predicting the ***same brain regions***, using the ***same settings and objective*** (i.e., intra-subject prediction). The results (R ± S.D) are shown below, with the best values in bold. **We find that our model still outperforms \[7,8\].**
>
> | Model | Pallidum | Caudate | Putamen | Accumbens | Average |
> | :---- | :---- | :---- | :---- | :---- | :---- |
> | Ours | **0.57 ± 0.18** | **0.58 ± 0.11** | **0.57 ± 0.16** | **0.60 ± 0.11** | **0.58 ± 0.02** |
> | \[7\] | 0.43 ± 0.15 | 0.49 ± 0.12 | 0.51 ± 0.14 | 0.43 ± 0.13 | 0.47 ± 0.04 |
> | \[8\] | 0.37 ± 0.04 | 0.47 ± 0.14 | 0.48 ± 0.16 | 0.42 ± 0.05 | 0.44 ± 0.05 |
>
>
> >**Point 4:**
> * **4.1)**: We apologize for the confusion, but our intention here for saying *"validate the robustness and applicability of our learned representations across neural data from different sites and scanners"* was to convey that our model shows promising applicability to another dataset, even if it is collected in a different condition (task) and on a different scanner (see **[Global response](https://openreview.net/forum?id=y6qhVtFG77&noteId=VLcizb9WHq)** and **Part 1** in the one-page PDF). We now notice that the word ”robustness” could cause confusion so we will omit this word in our revised manuscript. Evaluating the influence of noise requires simulating different noise levels, which is a bit out of the current scope of this paper, but it is a very interesting future direction.
>
> * **4.2)**: We had planned to provide further details on the building blocks of our model, including specific details of the multi-scale spectral representation learning module. Multiscale temporal representations and Short-Time-Fourier-Transform (STFT) are proving to be highly effective in representing neural data \[9,10\]. While the components of this module have well-established theoretical foundations in signal processing, our key contribution lies in integrating these components to work effectively together, resulting in a novel method and application (shown through ablation studies in *Section 4.4 Table 2*). Our planned additions include the following:
>
> 1. Formally describe (i.e. provide mathematical notations) how we utilized STFT to convert EEG time sequences into time-frequency features.
> 2. Discuss how the dimensions of frequency features depend on the time-window length with equations illustrating this relationship, and emphasize the rationale for selecting specific window lengths.
> 3. Introduce the trade-off between temporal and frequency resolution, further clarifying the reasoning behind using an approach that considers multiple window lengths.
>
> Thank you once again for your valuable comments. We hope our responses fully address your concerns.
>
>
> **References:**
>
> [1-6]: see refs. [1-6] in the Reviewer’s previous comment
>
> [7] Li et al. “Leveraging sinusoidal representation networks to predict fMRI signals from EEG”
>
> [8] Kovalev et al. “fMRI from EEG is only Deep Learning away: the use of interpretable DL to unravel EEG-fMRI relationships.” arXiv preprint.
>
> [9] Van De Ville et al. “When makes you unique: Temporality of the human brain fingerprint."
>
> [10] Samiee et al. “Epileptic seizure classification of EEG time-series using rational discrete short-time Fourier transform”

---

### Official Review · Reviewer_rcEv · 2024-07-12

**Soundness:** 3
**Presentation:** 3
**Contribution:** 4
**Rating:** 8
**Confidence:** 4

**Summary:**

The manuscript proposes an EGG-to-fMRI synthesis model. The framework implements a transformer architecture and uses a multi-channel feature combination expanded across the temporal axis. To evaluate the proposed model, EGG and fMRI data from 22 participants were recorded while they were in the resting state with eyes closed.

**Strengths:**

The manuscript addresses an interesting problem and can open opportunities for multimodal neuroimaging analysis. Overall, this line of investigation is little explored, therefore, the present manuscript is novel and of interest to the community.

The present manuscript is quite complete, (1) the model and rationale behind are sound; (2) a dataset is collected which allows a faithful evaluation of the proposed translation (from EEG to fMRI); (3) it's rather easy to read and follow the manuscript, (4) the reported results are promising.

**Weaknesses:**

The biggest weakness is that the framework and the dataset are only addressing the resting state. While this is an important baseline to investigate, it would have been great to explore the fidelity of the proposed framework when participants are presented with some stimuli.

It is unclear whether the source code and dataset will be released publicly.

The stability of the results is fully guaranteed given no statistical analyses are performed.

**Questions:**

What is "In-scan" in Table 1? This is not explained in the manuscript.

It is unclear to me why all methods are not evaluated for both inter-subject and in-scan. For example, isn't it possible to evaluate BIOT [44] for inter-subject data?

**Limitations:**

The manuscript sufficiently discusses its current limitations. I think the limitations section should tap into the particular scenario that the model has been evaluated on (resting state) and whether the results will be generalisable to other scenarios remains to be shown.

---

> ### Author Rebuttal · Authors · 2024-08-07
>
> We thank the reviewer for the thoughtful review and encouraging feedback on our manuscript! We are delighted that the reviewer found our work novel and of interest to the community. We are very encouraged by the reviewers’ evaluation on this work opening up opportunities for multimodal neuroimaging analysis. We greatly appreciate the positive comments on the soundness of our model, the clarity of our manuscript, and the promise of our results. We have addressed their specific questions below and included additional details in our global response.
>
> > **Evaluation on task fMRI:**
>
> Our motivation for focusing on resting-state fMRI data stems from the rich information that can be gained from naturally evolving brain dynamics. We sought to address the question of reconstructing resting-state fMRI signals since the spontaneous nature of the signals could present challenges beyond reconstructing block- or event-related fMRI task data.
> But we strongly agree with the reviewer that it is important to investigate the fidelity of our proposed model on fMRI data that contains a task. In this rebuttal, we now include an auditory task EEG-fMRI dataset for an in-depth evaluation, please kindly refer to the **global response part 1** for details.
>
> > **Data and code availability:**
>
> We will release all datasets on OSF, code on GitHub, and model weights on HuggingFace upon acceptance. This will provide the necessary resources for the community to validate, further explore, and build upon our models.
>
> > **Statistical analyses:**
>
> Please kindly refer to the global rebuttal response, where we provide additional results from our statistical analyses.
>
> > **Question 1:**
>
> “In-scan” stands for within-scan (i.e., intra-scan) prediction, where part of the scan is used for training and another part is used for testing. We clarified this terminology in the revised version.
>
> >**Question 2:**
>
> All methods can be applied to both inter-subject and in-scan predictions. We would like to clarify that we evaluated BIOT for both inter-subject and in-scan (intra-subject) frameworks (as shown in Table 1 in our original manuscript). For the in-scan evaluation, we originally selected the state-of-the-art EEG encoders (LaBraM [1] and BIOT [2]), as representative baselines for comparison. However, in our analysis for this rebuttal, we have also evaluated the intra-subject prediction performance on the two EEG-to-fMRI baselines (Li et al.[3] and BERIA [4]) to provide a more comprehensive analysis. Please see the comparison below, which shows the means and S.D. of correlation between prediction and ground truth ( full MSE table will be included in the revised manuscript). The significance of the paired t-test between our model and other baselines is indicated as follows: *:_p_<0.05; **: _p_<0.01; ***: _p_<0.001.
>
>
> | Model | Cu | He | MF | PA | Pu | Th | GLB |
> | :---- | :---- | :---- | :---- | :---- | :---- | :---- | :---- |
> | Ours | **0.59±0.17** | **0.57±0.18** | **0.50±0.17** | **0.56±0.14** | **0.44±0.18** | **0.48±0.21** | **0.59±0.16** |
> | LaBraM\[1\] | 0.54±0.18 | 0.52±0.20\*\* | 0.49±0.15 | 0.49±0.18\* | 0.41±0.18 | 0.45±0.18 | 0.49±0.17\*\* |
> | BIOT\[2\] | 0.53±0.22 | 0.52±0.21\* | 0.49±0.16\*\* | 0.46±0.11\*\*\* | 0.41±0.21 | 0.41±0.23 | 0.49±0.13\*\*\* |
> | Li et al.\[3\] | 0.46±0.22\*\* | 0.51±0.21 | 0.38±0.17\*\*\* | 0.46±0.20\* | 0.32±0.18\*\* | 0.40±0.19\* | 0.58±0.17 |
> | BERIA\[4\] | 0.36±0.24\*\*\* | 0.40±0.24\*\*\* | 0.29±0.23\*\*\* | 0.32±0.22\*\*\* | 0.23±0.19\*\*\* | 0.24±0.10\*\*\* | 0.46±0.24\*\* |
>
> > Cu: Cuneus; He: Heschl’s gyrus; MF: Middle Frontal Gyrus Anterior; PA: Precuneus Anterior; Pu: Putamen; Th: Thalamus; GLB: Global Signal
>
> Again, we genuinely appreciate the reviewer’s encouraging feedback and constructive suggestions! Please feel free to let us know if you have any further questions or comments.
>
> [1] Jiang, Wei-Bang, et al. "Large brain model for learning generic representations with tremendous EEG data in BCI." arXiv preprint arXiv:2405.18765 (2024).
>
> [2] Yang, Chaoqi, et al. "Biot: Biosignal transformer for cross-data learning in the wild." NeurIPS 36 (2024).
>
> [3] Li, Yamin, et al. "Leveraging sinusoidal representation networks to predict fMRI signals from EEG." Medical Imaging 2024: Image Processing. Vol. 12926. SPIE, 2024.
>
> [4] Kovalev, Alexander, et al. "fMRI from EEG is only Deep Learning away: the use of interpretable DL to unravel EEG-fMRI relationships." arXiv preprint arXiv:2211.02024 (2022).

---

> > ### Comment · Reviewer_rcEv · 2024-08-13
> >
> > I thank the authors for responding to my questions. I do not have any further questions.

---

> > > ### Author Response · Authors · 2024-08-14
> > > **Thank you!**
> > >
> > > Thank you once again for your valuable and encouraging feedback! We sincerely appreciate the time and effort you put into the review process.

---

### Official Review · Reviewer_EkS2 · 2024-07-19

**Soundness:** 3
**Presentation:** 3
**Contribution:** 3
**Rating:** 5
**Confidence:** 5

**Summary:**

In this work, the authors present a deep learning architecture for inferring functional magnetic resonance imaging (fMRI) signal from electroencephalography (EEG) data. The proposed model, named NeuroBOLT, utilizes transformer backbones to provide spatial, temporal, and frequency-based features from the EEG are utilized for reconstruction. The authors demonstrate the performance of their architecture on a small (N=22) data set using a propriatary data set of simultaneously measured EEG-fMRI.

**Strengths:**

fMRI reconstruction from simultaneous EEG is a fascinating topic, and a difficult problem to tackle. The approach taken by the authors in this work is novel for the task at hand, i.e. using a multi-scale spectral feature embedding. Although the decision to use multi-scale spectral embeddings is not new in MRI analysis, as far as I could find the approach has not been utilized for this particular problem and the authors address novel problems for their application to simultaneous EEG-fMRI data in a deep learning architecture. At best this paper is a novel methodological tweak applied with state of the art architectures to see improvements over other deep learning baselines.

The breadth of the experiments attempted by the authors is promising; however, see my discussion below for more of a discussion of the limitations of the experiments performed.

The authors also perform an ablation study to explore how the inclusion of Multi-Scale Spectral features improves model performance, thus demonstrating the benefit of combining the multi-scale spectral features with the spatiotemporal. This is well appreciated.

**Weaknesses:**

The major weaknesses of this work come down to weaknesses in the empirical evaluation. I am afraid that in its current state, the evaluation does not lead to a convincing demonstration of this method for fMRI reconstruciton, and the claims in the introduction about novelty coming from the application to multiple brain regions and resting-state fMRI seem somewhat overemphasized. Currently this brings it to a full reject as the paper is otherwise sound but the limitations in the evaluation are significant enough to bring it well below the threshold, and cannot be easily addressed in the rebuttle I believe.

First, I will highlight the lack of reported standard deviations or error bars in any of the results. No standard deviations are provided in tables 1 or 2, or in any of the figures providing results. In the checklist, the authors state "Error bars are not reported at the current stage because it would be too computationally expensive to compute over all the brain regions and for all participants,also there is limited space in the paper to put all the statistics. But we could always add this information if reviewers think it’s important to know."

While I appreciate the authors' acknowledgement of this exclusion, I do think error bars are absolutely necessary to demonstrate the efficacy of the proposed method. The demonstrated improvements are often quite small (e.g. improvement from 0.540 to 0.588 in table 1), and it is not clear if the purported improvements can be explained away from model noise. I could not find any information about controlling model initialization or seeds as well to ensure that random initializations played a less significant role between experiments even with the same architecture on different regions. I absolutely think error bars are necessary for this work, and the reasoning provided by the authors is not mitigated elsewhere or behind a more significant barriers other than training and evaluation time. Additionally, the authors could have mentioned this omission in the limitation section of their main paper since I had to go to the checklist to be sure the authors were aware of the issue.

Second, in the abstract the authors highlight the ability to "generalize to other brain areas" and "other conditions (such as resting state)".  Unless I am missing something, I cannot find any experiments by the authors that address these particular gaps. The authors do provide inter- and intra-subject predictions which is interesting; however, their model is still only trained on individual ROIs, and they don't include any experiments demonstrating transfer learning between models trained on other regions, and they do not include any experiments studying other tasks BEYOND resting-state fMRI. Thus, the paper falls into the same limitation as past works which were only focused on task, just in the other direction. This work would have been much more compelling if they could demonstrate a model which trained well both on task and rest related data, or even better, which could reconstruct task-related data despite only being trained on resting-state fMRI. The acknowledgement of the limitations in the literature is thus misleading as the proposed method still suffers from these same limitations. The choice to only demonstrate the results for several ROIs highlights this limitation - it would be okay if the authors did not seem to imply elsewhere that their model gets around the single-ROI training approach from past methods.

Clearly, the N in this study is quite small. This is to be expected as simultaneous EEG-fMRI is still quite rare as a sequence to collect; however, the authors seem to gloss over all of the myriad issues which will come with training their data over such a small data set. I am not penalizing this work for the small N in and of itself, but as I cannot find any mention of common obstacles such as overfitting, bias towards particular kinds of reconstruction errors, and other limitations that would inevitably arise. I am extremely surprised I could find no mention of pretraining anywhere in this work, which I almost imagine would be hugely necessary for these kinds of studies with very small data sets. Again, it's not necessarily a limitation in and of itself to not do these things, with how the paper currently reads these seem to be touted as benefits of the new approach which are not backed up by evidence.

**Questions:**

How well does the model trained on other ROIs transfer to reconstruction of completely different ROIs?

How might scanner model and particular parameters of the resting state sequence affect reconstruction?

Why do you not compare anywhere with Source Localization? Source Localization is only mentioned once offhand, and its limitations and efficacy as a reconstruction technique are not gone into in detail. I am surprised it was not included as a baseline method in fact.

**Limitations:**

The authors do provide some discussion of the limitations of their work; however, as I have noted above, there are some limitations which are ommitted from the main body of text which at least should have been acknowledged in this section.

The authors state they have IRB approval in section 4.1 of their paper. I see no reason for additional review.

---

> ### Author Rebuttal · Authors · 2024-08-07
>
> We truly appreciate your excellent suggestions! We address specific concerns below. Please see the PDF in the general response for additional details and figures.
>
> > **Evaluation on task fMRI**
>
> Our motivation for focusing on resting-state fMRI stems from the rich information that can be gained from naturally evolving brain dynamics. We sought to address the question of reconstructing resting-state fMRI signals since the spontaneous nature of the signals could present challenges beyond reconstructing block- or event-related fMRI task data.
> But we strongly agree that it would be more compelling if our proposed model is evaluated on task data, in addition to rest. In this rebuttal, we now include an auditory task EEG-fMRI dataset for additional evaluation, please kindly refer to the global response part 1 for details.
>
> > **Lack of error bars/standard deviations**
>
> We have now revised our work to include error bars in figures and standard deviations in tables, and statistical significance. Please see the global response and the PDF attachment for details.
>
> > **Model initialization and seeds**
>
> To ensure consistency during model training, we set a fixed seed. This clarification has been added to our revised manuscript. While this is certainly an important robustness test, we plan to focus on this point in a future study due to the large scope of the current analyses. We will raise this important point in the Discussion of the revised manuscript.
>
> > **Single-ROI training**
>
> We now realize that the wording "generalize to other brain areas" can imply that a single trained model can predict fMRI signals from different brain areas. In fact, we had meant to convey that our modeling framework can be trained to reconstruct the fMRI signal from an arbitrary brain region (but, indeed, ROI-specific models are trained). We have now clarified the wording in an effort to avoid this misunderstanding.
>
> > **Small N**
>
> We acknowledge the small sample size and agree that it is important to make this point clear to readers. We have carefully reviewed the entire manuscript and made sure this limitation is discussed more clearly in the main text.
>
> Regarding pretraining, we would like to clarify that we do leverage the pre-trained weights from the EEG foundation model (LaBraM [1], trained on 2500 hours of various types of EEG data from around 20 datasets) for our “Spatiotemporal Representation Learning module.” In the original manuscript, we had briefly discussed pretraining in the last paragraph of the introduction section and in section B.1 of the appendix. We have now added clarifications across the manuscript (specifically in the Methods and in Section 4, Experiments: Implementation Detail), describing the pre-training procedure and its importance.
>
> Moreover, the additional results on the task-based fMRI collected at a different site demonstrate that training our model on a relatively small sample of resting-state data shows promising potential to generalize to different task conditions and hardware.
>
> [1] Jiang, Wei-Bang, et al. "Large brain model for learning generic representations with tremendous EEG data in BCI." arXiv preprint arXiv:2405.18765 (2024).
>
> > **Question 1**
>
> Please also see our response to “Single-ROI training” above. The current framework takes multi-channel EEG as input and predicts the fMRI ROI signal from the region on which the model is trained. We would therefore expect the model trained for one ROI would tend to reconstruct other fMRI ROIs if they share correlated temporal fluctuations. In future work, we will extend the model to predict multiple fMRI ROIs signals at once. We are revising the main text accordingly.
>
> > **Question 2**
>
> While our new task analysis presents an initial probe into this question, showing promising performance on task data acquired with different hardware/parameters, answering this question directly would require future work that systematically investigates altering scan parameters (while keeping the task condition, and ideally the subject, consistent) and would involve collecting more data. We believe this is a useful avenue for future work.
>
> > **Question 3**
>
> This is a great point! Source localization is indeed a valuable method. However, we did not include source localization for the following reasons:
>
> **(1) Limited number of EEG electrodes:** Our EEG data consist of 32 electrodes. After excluding the ECG, EOG, and EMG channels from the model input, this is reduced to 26 channels. However, larger numbers of channels (e.g., >64) may be needed for sufficiently accurate source localization [1-2].
>
> **(2) Hemodynamic filter between source electrical signal and fMRI signal:** Even if we can map the EEG into source space, the output signals would be neural electrical signals. Our present study focuses on predicting fMRI hemodynamic signals, which are blurred/delayed relative to electrical signals. Although fMRI signals can be estimated by convolving neural signals with the hemodynamic response function (HRF), the shape and peak of the HRF vary across different brain regions and individuals, which would pose challenges for aligning between fMRI and source-reconstructed EEG signals.
>
> However, we do think that source localization is an excellent future direction.
>
> [1] Michel, Christoph M., and Denis Brunet. "EEG source imaging: a practical review of the analysis steps." Frontiers in neurology 10 (2019): 325.
>
> [2] Michel, Christoph M., et al. "EEG source imaging." Clinical neurophysiology 115.10 (2004): 2195-2222.
>
> We genuinely appreciate your commitment to ensuring the rigor of our paper and the constructive suggestions. These insights have significantly improved this work. We hope our rebuttal addresses your concerns and please let us know if you have any additional questions/comments!

---

> ### Comment · Reviewer_EkS2 · 2024-08-09
> **Score Adjustment**
>
> The rebuttal provided by the authors addresses much of the concerns raised here. The requested error bars somewhat lower the impact of the model compared to baselines, as there is high variance and many baselines perform fairly comparably; however, there is demonstrative improvement. I have raised my score to a borderline accept.

---

> > ### Author Response · Authors · 2024-08-13
> > **Thank you for increasing your score**
> >
> > Thank you so much for your prompt response and raising the score. We are happy that our additional experiments have improved this work, and sincerely appreciate your feedback, which helps to improve the quality of our paper.
> >
> > We acknowledge that the error bars are relatively large across all methods. While variability in the performance across individuals presents a challenge, we find it promising that our method achieves consistently higher mean performance than the baselines for all (or all but one) of the ROIs in the intra/inter-scan comparisons (Figs. R1.2, R2.1, R2.2 in the one-page PDF), though indeed not all of these differences are statistically significant. By contrast, considering only the baseline methods, these appear to vary across ROIs in terms of which model achieves the highest mean performance. Additionally, we believe this is the first study to incorporate and adapt these state-of-the-art EEG Encoding baselines in this specific application of EEG-to-fMRI translation.
> >
> > We believe that your comment is highly valuable, and will include a discussion of this limitation in our revised manuscript. In addition, we are planning further in-depth exploration of the factors that may drive performance variability across individuals in our future work, as this is indeed an important issue.
> >
> > Thank you once again for your valuable comments!

---

### Author Rebuttal · Authors · 2024-08-07

We thank all the reviewers for their time and their insightful, constructive suggestions. We are excited that all reviewers find the topic of our paper important and fascinating. Reviewers found our study to be novel and well-motivated with promising results, and also noted that our manuscript is well-organized (Reviewer bwvZ) and highly readable (Reviewers rcEv, bwvZ).

We have addressed below two main concerns that are raised by the reviewers: absence of (1) evaluation on task-related fMRI, and of tests regarding generalizability of our model from resting-state to task-related data; and (2) statistical measures to demonstrate the model's efficacy and feasibility. In addition to this global response, we include point-by-point responses addressing the comments of each reviewer in the separate response sections below.

> #### **1. Generalization to task-related fMRI**

One concern that the reviewers shared involved the lack of task fMRI data in our experiments. Accordingly, we have now run experiments using **a simultaneous EEG-fMRI dataset collected during auditory tasks** (see details below **\*\*Auditory Task**). We conducted additional experiments using this dataset: **(1) zero-shot generalization,** where we pre-trained our model on resting-state fMRI data and evaluated the performance on task fMRI data; **(2) intra-subject prediction and inter-subject prediction,** where we trained and evaluated models using only task fMRI data; **(3) fine-tuning,** where models trained on resting-state data were fine-tuned with task fMRI data, and **(4) joint-training,** where we jointly trained (using both resting-state and auditory task fMRI data) and evaluated the model on the respective held-out test sets of both datasets. For these experiments, we used the same parameter settings for the model training and optimization as in the main paper. Specifically, in these experiments we addressed:


*  **Feasibility of our model framework on task fMRI (new experiments 2\)**: In the PDF attachment, Fig R1.1 shows the **intra-subject** prediction performance distribution, with average correlation \= 0.51±0.18 across 7 ROIs and 16 scans. Fig R1.2 displays the **inter-subject**, unseen whole-scan prediction (trained on 3+6 scans, validated on 1+2 scans, tested on 2+2 scans for Fast+Sparse auditory tasks) performance compared with other baselines. Our model exhibits superior performance across 6 ROIs.
* **Generalizability of our model trained with resting-state data (new experiments 1, 3 and 4).** The performance of our pretrained model on *zero-shot* whole-scan task fMRI reconstruction (see in Table R1, the first row of data) achieved performances comparable to that of our original resting-state data, with even better performance in several regions compared with the model that was trained only on task fMRI. With further *fine-tuning* on the task dataset, the performance improved significantly. Moreover, carrying out *joint training* using both resting-state and auditory task fMRI datasets resulted in the best performance across 4 ROIs in task fMRI prediction. Our results also suggest that joint training is not necessarily facilitating the prediction on resting-state fMRI (beyond training on resting-state data alone), which might be due to the smaller sample size of task data and richer variability of brain dynamics in the resting state.


The results from the above experiments demonstrate initial evidence that our model is indeed able to generalize across resting-state and task fMRI data, as well as across different sites and hardware settings.


> #### **2. Statistical measures**

Another concern raised by the reviewers is the lack of statistical measures to demonstrate the model's efficacy and feasibility. To address this, we added bar plots (with error bars showing S.D.) and indicated the statistical significance for resting-state intra- and inter-subject predictions. For statistical testing, the correlation values are firstly Fisher-z transformed and a paired t-test was employed to test the significance of improvement of our model compared with other baselines (\*: _p_-value\<0.05; \*\*: _p_-value\<0.01; \*\*\*: _p_-value\<0.001). Our task fMRI results shown in the PDF attachment also include the S.D. and error bars, and the statistical significance of the baseline comparisons.

> **3. Data and code availability.**

We will release all datasets on OSF, code on GitHub, and model weights on HuggingFace upon acceptance. This will provide the necessary resources for the community to validate, further explore, and build upon our models.

Finally, we would like to express our deep appreciation for the reviewers' comments and their recognition of the promising potential of our work to inspire future advancements in multi-modal neuroimaging. All the additional results (experiments on task fMRI, R values, and MSE values with S.D.) will be properly incorporated into our revised manuscript, with detailed information about task-fMRI data collection/preprocessing and implementation details of the additional experiments.


> \*\* **Auditory Task details.** Specifically, binaural tones were delivered with randomized inter-stimulus intervals (ISI), and there are two versions of the task that differed only in the timing of tone delivery: (1) **a fast ISI version** (6 scans, 500 TR/scan, ISI mean(SD) \= 5.6(0.7) sec, TR \= 2.1 sec), (2) **sparse ISI version** (10 scans, 693 TR/scan, ISI mean(SD) \= 40.1(14.6) sec, TR \= 2.1 sec). Subjects were asked to keep their eyes closed the entire time and to make a right-handed button press as soon as possible upon hearing a tone. This dataset was collected at a different site, different MR scanner (3T Siemens Prisma scanner) and using a slightly different EEG cap (32 channels but with partially different electrode settings). Detailed information about data collection and preprocessing will be included in the revised version of the manuscript.

---

### Decision · Program_Chairs · 2024-09-25

**Decision:**

Accept (poster)

**Comment:**

The paper introduces a model that adopts the transformer architecture to convert EEG data into fMRI. The model leverages spatial, temporal, and frequency-based features to reconstruct the fMRI signal. In our review process, while some considered the methodological advancements incremental, the majority of reviewers pointed out the significance of a thorough empirical evaluation on a simultaneous EEG/fMRI dataset. All reviewers recognized the importance of the tackled problem and the strong general interest of the field in it.

Addressing specific concerns, reviewer bwvZ was mostly satisfied apart from the issue of citing the relevant literature. This remaining concern was addressed in the rebuttal, with the proposed inclusion of the relevant literature in the final version of the manuscript. I disagree with the reviewer on the basis that this alone is a strong enough reason to reject the paper. The reviewer unfortunately failed to provide specific details regarding which "relevant literature" was omitted.

In summary, this paper displays promising features despite the following three areas of concern:

- A high variance in the results of empirical evaluation, which suggests the proposed advances may not be as groundbreaking as compared to existing methodologies.
- An evaluation performed on a proprietary dataset, eliciting potential concerns over reproducibility.
- A perceived limited innovation in methodology.

The significance of the problem addressed and the empirical results that convinced two out of three reviewers make this paper worthy of acceptance. However, due to the aforementioned limitations, it is more suitable to be presented as a poster. This gives the authors an opportunity to discuss these concerns with the wider research community, potentially strengthening their work in the process.